# Genome-Wide Identification and Expression Analyses of the Thaumatin-Like Protein Gene Family in *Tetragonia tetragonoides* (Pall.) Kuntze Reveal Their Functions in Abiotic Stress Responses

**DOI:** 10.3390/plants13172355

**Published:** 2024-08-23

**Authors:** Zengwang Huang, Qianqian Ding, Zhengfeng Wang, Shuguang Jian, Mei Zhang

**Affiliations:** 1Guangdong Provincial Key Laboratory of Applied Botany, South China National Botanical Garden, Chinese Academy of Sciences, Guangzhou 510650, China; huangzengwang23@scbg.ac.cn (Z.H.); dingqianqian@scbg.ac.cn (Q.D.); wzf@scbg.ac.cn (Z.W.); jiansg@scbg.ac.cn (S.J.); 2Guangdong Provincial Key Laboratory of South China Agricultural Plant Molecular Analysis and Genetic Improvement, South China National Botanical Garden, Chinese Academy of Sciences, Guangzhou 510650, China; 3University of Chinese Academy of Sciences, Beijing 100039, China; 4Key Laboratory of Vegetation Restoration and Management of Degraded Ecosystems, South China National Botanical Garden, Chinese Academy of Sciences, Guangzhou 510650, China; 5Key Laboratory of National Forestry and Grassland Administration on Plant Conservation and Utilization in Southern China, South China National Botanical Garden, Chinese Academy of Sciences, Guangzhou 510650, China; 6CAS Engineering Laboratory for Vegetation Ecosystem Restoration on Islands and Coastal Zones, South China National Botanical Garden, Chinese Academy of Sciences, Guangzhou 510650, China

**Keywords:** thaumatin-like proteins (TLPs), abiotic stress, *Tetragonia tetragonoides* (Pall.) Kuntze

## Abstract

Thaumatin-like proteins (TLPs), including osmotins, are multifunctional proteins related to plant biotic and abiotic stress responses. *TLP*s are often present as large multigene families. *Tetragonia tetragonoides* (Pall.) Kuntze (Aizoaceae, 2*n* = 2x = 32), a vegetable used in both food and medicine, is a halophyte that is widely distributed in the coastal areas of the tropics and subtropics. Saline–alkaline soils and drought are two major abiotic stress factors significantly affecting the distribution of tropical coastal plants. The expression of stress resistance genes would help to alleviate the cellular damage caused by abiotic stress factors such as high temperature, salinity–alkalinity, and drought. This study aimed to better understand the functions of *TLP*s in the natural ecological adaptability of *T. tetragonoides* to harsh habitats. In the present study, we used bioinformatics approaches to identify 37 *TtTLP* genes as gene family members in the *T. tetragonoides* genome, with the purpose of understanding their roles in different developmental processes and the adaptation to harsh growth conditions in tropical coral regions. All of the *TtTLP*s were irregularly distributed across 32 chromosomes, and these gene family members were examined for conserved motifs of their coding proteins and gene structure. Expression analysis based on RNA sequencing and subsequent qRT-PCR showed that the transcripts of some *TtTLP*s were decreased or accumulated with tissue specificity, and under environmental stress challenges, multiple *TtTLP*s exhibited changeable expression patterns at short (2 h), long (48 h), or both stages. The expression pattern changes in TtTLPs provided a more comprehensive overview of this gene family being involved in multiple abiotic stress responses. Furthermore, several *TtTLP* genes were cloned and functionally identified using the yeast expression system. These findings not only increase our understanding of the role that *TLP*s play in mediating halophyte adaptation to extreme environments but also improve our knowledge of plant TLP evolution. This study also provides a basis and reference for future research on the roles of plant *TLP*s in stress tolerance and ecological environment suitability.

## 1. Introduction

*Tetragonia tetragonoides* (Pall.) Kuntze, also called New Zealand spinach or French spinach, is a calcium-rich vegetable with high commercial values and widely distributed in the coastal regions of the tropics and subtropics. *Tetragonia tetragonoides* is a halophyte and can be planted near the beach, on islands, and on big reefs for ecological restoration or vegetable garden construction. This species is highly adapted to seawater and drought conditions [1] and therefore becomes the pioneer species for marine agriculture [2]. Additionally, *T. tetragonoides* is also a medicinal plant, as its extracts are used to treat gastrointestinal diseases owing to its antioxidant, antidiabetic, and anti-inflammatory effects [3]. Soil salinization/alkalization has become a serious threat that limits plant growth, development, and distribution and is an increasingly severe global environmental issue due to the combination of natural environmental changes and human activities [4,5]. For tropical and subtropical coastal plants, their morphology has changed with the environment, including leaf succulents and salt bladders in epidermal cells, especially under extreme salinity and drought stress challenges [3,5]. To deal with these external constraints, plants have also evolved highly complex and sophisticated response mechanisms to avoid damage from drought, salinity toxicity, extreme temperatures, chemical toxicity, oxidative stress, and subsequent physiological water imbalances in the osmotic potential of cells. The related molecular mechanisms are usually accompanied by the accumulation of stress-related proteins, which often include protein kinases, transcription factors (TFs), reactive oxygen species (ROS)-scavenging or ROS-detoxification proteins, channel proteins, and some molecular chaperones [6].

The thaumatin-like protein (TLP) family, also known as pathogenesis-related protein family 5 (PR-5) or osmotin protein, is a group of low-molecular-weight (20–26 kDa) proteins with 16 conserved cysteine (Cys) residues [7,8]. The expression of plant *TLP*s is induced by multiple abiotic stressors, such as high temperature, cold, salt, and drought stress. In addition, as a group of pathogenesis-related genes, *TLP*s are also induced by biotic stressors, including pest- and disease-dependent stressors [7,8,9]. Many studies have been performed to determine the role of TLPs in plants due to their osmo-protective and antifungal properties [10,11].

Recently, an increasing number of studies have reported that plant *TLP*s could provide significant elevated tolerance against abiotic stressors when overexpressed in microorganisms or plants. *Solanum nigrum* is a solanaceous weed, and *SnOLP* overexpression in soybeans (under the control of the Arabidopsis *UBQ3* promoter) confers enhanced drought tolerance [12]. An *Ocimum basilicum PR-5* family member (*ObTLP1*) shows a methyl jasmonate (MeJA)-responsive expression pattern and presents biotic/abiotic stress responses or multiple phytohormone elicitations. The ectopic expression of *ObTLP1* in Arabidopsis leads to enhanced tolerance to infection by two phytopathogenic fungi, as well as to dehydration and salt stress [7]. Overexpression of the sesame (*Solanum nigrum*) osmotin gene *SindOLP* improves drought, salt, oxidative stress, and disease tolerance through altered biochemical parameters and reduced reactive oxygen species (ROS) accumulation in sesame plants [13]. *Tripogon loliiformis* is an Australian native resurrection grass with a rapid response to water deficits and quick recovery upon rehydration. The *T*. *loliiformis* osmotin gene *TlOsm* was isolated from a drought-induced cDNA library, and its expression in vivo was significantly induced by cold, drought, and salinity stress, indicating that *TlOsm* is involved in multiple abiotic stress responses. Further transgenic assays showed that *TlOsm* overexpression in rice could cause enhanced tolerance of rice plants to cold, drought, and salinity stress [14]. The rice genome contains 40 *osmotin* members, and *OsOLP1* encodes a secreted protein. Its expression results in multiple changes under various stressors and has been associated with the desiccation/dehydration stress response. Overexpressing *OsOLP1* in rice results in high drought tolerance, while *OsOLP1* knockout lines have a severely reduced abscisic acid (ABA) content, decreased lignin deposition, and weakened drought tolerance [15]. Overexpression of the cotton TLP gene *GhTLP19* in Arabidopsis results in higher tolerance to drought than that of control plants, while virus-induced gene silencing (VIGS) of this gene in cotton results in an insensitive phenotype to drought and phytopathogenic fungi *Verticillium dahliae* [16]. *Triticum aestivum* contains 93 *TLP* family members, and the expression patterns of *TaTLP*s indicated this family was possibly involved in wheat development processes and abiotic or biotic stress responses. In addition, heterogeneous expression of *TaTLP2-B* in yeast provides improved tolerance to cold, heat, osmotic, and salt stresses [17]. *Ammopiptanthus nanus* is tolerant of cold and is a rare evergreen broad-leaved shrub distributed in the temperate zone of Central Asia. The expression of *AnTLP* family genes is highly related to the environmental cold stress response, and over-expression of *AnTLP13* in *Escherichia coli*, yeast cells, and tobacco leaves enhances its cold stress tolerance [18]. However, very few studies have indicated that the accumulation of osmotin-like proteins might lead to reduced osmotic stress tolerance in plants, such as *HbOsmotin* in *Hevea brasiliensis* [19].

Genes encoding osmotins and TLPs have been identified in diverse plant species and play multiple functions in mediating stress tolerance responses [8,9,10,11]. The poplar *TLP* gene family is likely involved in leaf rust resistance and organ development [20]. The grape (*Vitis vinifera* L.) *TLP* gene family contains 33 putative members. The transcripts of several *VvTLP*s accumulate specifically after pathogen inoculation, and *VvTLP29* overexpression in *Arabidopsis thaliana* enhances its resistance to powdery mildew and the bacterium *Pseudomonas syringae* pv. tomato DC3000 but decreases resistance to *Botrytis cinerea* [21]. The barley *TLP* gene family is suspected to be involved in seed germination and the malting process [22]. Melon (*Cucumis melo*), cotton (*Gossypium barbadense*), strawberry (*Fragaria* × *ananassa*, 2*n* = 8x = 56), and garlic (*Allium sativum*) genomes contain 29, 90, 76, and 32 *TLP*s, respectively [23,24,25,26], and they all play important roles in developmental processes and diverse stress condition responses, especially in disease resistance.

Since many plant TLPs have been proven to provide osmotic adjustment during salinity and moisture stress [8,9,10,11], and because *T. tetragonoides* is a halophyte and mostly challenged by saline–alkaline, arid, and semi-arid conditions due to its native habitats, we proposed that these proteins have significant protective roles for the growth of *T. tetragonoides*. The aim of this study was to characterize the *TLP*s in *T. tetragonoides*, one of the most promising tropical seawater vegetables, and to further determine whether *TtTLP* genes were involved in the response and adaptation of this species to high salinity/alkalinity and seasonal drought stress in its special native habitats, mainly including tropical coastal regions and reefs/islands. The detailed characterization of plant *TLP*s focuses more on the defense responses, and our research here provides important data for understanding the biological function and abiotic stress responses of *TtTLP*s in *T. tetragonoides*.

## 2. Results

### 2.1. Identification of the T. tetragonoides TLP Family

Thirty-seven *TLP* genes have been identified from the *T. tetragonoides* genome using InterProscan search combined with BLAST confirmation for their coding proteins (Table 1; Appendix A). The TtTLPs had a conserved TLP domain on their C-terminus, and according to their gene loci on the chromosomes, these genes were designated as *TtTLP1*–*37*. Among the 16 pairs of chromosomes in *T. tetragonoides*, chromosomes 05, 08, and 12 held no *TtTLP*s. Chromosomes 04 and 06 had the most *TtTLP*s (five genes), while chromosomes 07 and 14 both had four *TtTLP*s. Chromosomes 02 and 16 both had three *TtTLP*s, and chromosomes 01, 03, 09, 11, 13, and 15 each contained two *TtTLP*s. There was only one *TtTLP* (*TtTLP24*) on chromosome 10 (Figure 1). In general, three pairs of modules for *TtTLP*s with adjacent gene localization, namely *TtTLP10*/*TtTLP11*/*TtTLP12* and *TtTLP15*/*TtTLP16*/*TtTLP17*, *TtTLP20*/*TtTLP21* and *TtTLP27*/*TtTLP28*, and *TtTLP22*/*TtTLP23* and *TtTLP33*/*TtTLP34* (Figure 1), could be evolutionarily close, and this evolutionary relationship could also be deduced from the gene loci (Table 1) and phylogenetic relationship (Figure 2).

### 2.2. Evolutionary Characterization of TLP Proteins and TtTLP Gene Structures

To explore the molecular phylogeny between TtTLPs, we performed phylogenetic analysis and established unrooted phylogenetic trees (Figure 2). Due to proteins’ conserved motifs being critical for their biochemical functions, motif analyses were also carried out using the MEME tool with ten motifs (Figure 2). The majority of the TtTLPs contained a conserved Thaumatin_2 domain (PS51367, InterPro analysis), with the exception of TtTLP7 and TtTLP36 with an atypical TLP domain (Appendix A). This agreed with the MEME discovery prediction for TtTLPs (Figure 2).

To better understand the structural features of the *TtTLP* genes, exon–intron framework analyses were performed using the GSDS 2.0 tool. The number of introns in the 37 *TtTLP*s ranged from 0 to 5, of which 10 *TtTLP* genes (*TtTLP1*, *TtTLP2*, *TtTLP6*, *TtTLP10*, *TtTLP11*, *TtTLP12*, *TtTLP15*, *TtTLP16*, *TtTLP17*, and *TtTLP19*) contained no introns. Twelve *TtTLP* genes (*TtTLP3*, *TtTLP7*, *TtTLP8*, *TtTLP9*, *TtTLP14*, *TtTLP23*, *TtTLP25*, *TtTLP26*, *TtTLP29*, *TtTLP31*, *TtTLP33*, and *TtTLP36*) contained one intron, and eleven *TtTLP* genes (*TtTLP4*, *TtTLP5*, *TtTLP13*, *TtTLP18*, *TtTLP21*, *TtTLP22*, *TtTLP27*, *TtTLP28*, *TtTLP32*, *TtTLP34*, and *TtTLP35*) contained two introns. Only two *TtTLP* genes (*TtTLP30* and *TtTLP37*) contained three introns, and only *TtTLP20* and *TtTLP24* contained five introns. No *TtTLPs* contained four introns (Figure 3). When compared with the phylogenetic clustering analysis of TtTLPs and protein motif prediction (Figure 2), the exon–intron structure of *TtTLP* genes in the same subgroup was unrelated to the protein motif analysis.

To gain insight into the relationship between TtTLPs and those of other plant species, we used data from 58 TLPs in the model plants *Arabidopsis* (27 AtTLPs) and rice (31 OsTLPs) [23]. These sequences were then aligned, and the phylogenetic tree was constructed via the NJ method (Figure 4). The resulting tree classified these TLPs into 10 subgroups, named groups 1 to 10, based on their phylogenetic relationship. These results are similar to those observed in melon (*Cucumis melo* L.) CmTLPs [23]. In each subgroup, there were 2–6 TtTLPs, indicating that based on the conserved domains combined with functional analysis, the existence of different TtTLPs with specialized functions was also relatively average.

### 2.3. Duplication Event Investigation

The emergence and existence of the gene members in a large gene family usually include a single gene and several gene duplication patterns: whole-genome duplication (WGD), tandem duplication (TD), proximal duplication (PD), transposed duplication (TRD), and dispersed duplication (DD). Duplication event analyses were carried out to understand the role of these events in the evolution and expansion of the *TtTLP* gene family in this special habitat species. Most *TtTLPs* were generated by WGD, and only five *TtTLPs*, namely *TtTLP5*, *TtTLP18*, *TtTLP19*, *TtTLP24*, and *TtTLP31*, were generated by DD. Two *TtTLPs*, *TtTLP12* and *TtTLP17*, were generated by PD, and only *TtTLP11* had a TD pattern (Appendix A). All duplicated gene modules had a similar aggregated distribution on the chromosomes (Figure 1).

Throughout evolution, evolutionary forces and natural pressures inevitably affected the duplicated genes with different patterns. To understand the evolutionary divergence between the paralogous gene pairs, Ka/Ks analysis was carried out in the *TtTLP* gene family. A Ka/Ks ratio of more than 1 (Ka/Ks > 1) suggests positive (non-purifying), and a ratio less than 1 (Ka/Ks < 1) indicates negative (purifying) selection pressure. A ratio equal to 1 (Ka/Ks = 1) indicates neutral selection. All paralogous genes showed a Ka/Ks ratio less than 1 (Ka/Ks < 1), suggesting negative or purifying selection on duplicated TLP genes (Appendix A).

### 2.4. Features of TtTLP Proteins

After verification of the complete ORF of the full-length cDNA of each *TtTLP*, the predicted amino acid (aa) sequences of the TtTLPs were calculated using biological programs. Among the 37 TtTLPs identified, TtTLP30 was the longest, with 825 aa, and TtTLP8 was the shortest (179 aa). Most of the TtTLPs had a length between 200 and 400 aa. The molecular weights (MW) of these TLPs ranged from 19.33 to 91.27 kD, with isoelectric point (pI) values between 4.23 and 9.28. Detailed information on the TtTLPs, including their names, gene loci, protein lengths (aa), MWs, major amino acid contents, pI, instability index (II), aliphatic index (AI), grand average of hydropathicity (GRAVY) values, disordered aa contents, and topology characteristics, is summarized in Table 1. The subcellular localization for all TtTLPs predicted by WoLF_PSORT and Plant-PLoc presented slightly different results, while most TtTLPs were localized extracellularly.

The conserved cysteine residues in plant TLPs are typical features of the THAUMATIN_2 domain, and the REDDD motif (arginine, glutamic acid, and three aspartic acid residues) is also known to confer antifungal activity to plant TLPs [27]. Here, we also searched for these amino acid residues in all TtTLPs, and the presence of these conserved amino acids described for the thaumatin domain in the aligned sequences of the TtTLPs further ensures the conservation of the biochemical functions of TtTLPs.

### 2.5. Cis-Acting Elements (CEs) of the TtTLP Promoter Sequences

The CEs in the gene promoter region are necessary for the regulation of gene expression during plant development and under different environmental conditions, mainly by binding to specific TFs. The promoter sequences in the 2 kb region upstream of all *TtTLP*s’ 5′-untranslated regions (5′-UTRs) were analyzed. Based on their putative functions, the 13 identified CEs were segregated into two groups: hormone-specific and abiotic stress-responsive elements. Here, we mainly summarized their numbers (Figure 5A) and localized the specific CE positions (Figure 5B) in the promoter regions. This information explained the biological functions of *TtTLP*s by predicting their possible expression regulatory mechanisms.

The results of the *TtTLP* CE analysis are shown in Figure 5 and Appendix A. Most *TtTLP*s (28 of 37) had CEs responsive to ABA (ABRE), and all *TtTLP*s had elements responsive to hormones, most of which responded to at least one of the six hormones searched. Anaerobic-responsive elements were also relatively common in *TtTLP* promoters, indicating their possible roles for water logging or sand/soil burial. More than half of *TtTLP*s had abiotic stress-inducibility elements, such as as-1, HSE, LTRE, and TC-rich repeats. Therefore, the expression of *TtTLP* genes may be regulated by CEs associated with abiotic stress responses. MYB- and MYC-binding sites (MYB and MYC) were also observed. In particular, only the *TtTLP16* promoter region did not contain a MYB-binding site, and the other 36 *TtTLP* promoters held at least 1 MYB-binding CE (up to 15 in *TtTLP1* and *TtTLP25* promoters). Similarly, MYC-binding CEs commonly exist in all *TtTLP* promoters (Appendix A). In plants, MYB and MYC TFs are not only about abiotic stress responses but are also related to plant development processes [28,29]. The presence of MYB- and MYC-binding CEs in all *TtTLP* promoters suggests that further possible functions of the *TtTLP* family are involved in *T. tetragonoides* development and environmental adaptation.

### 2.6. Expression Profiles of TtTLPs in Different Tissues and Plants in Response to Stress

Tissue-specific expression profiles of *TtTLP*s were analyzed using RNA-seq in the roots, stems, leaves, flower buds, and young fruit of *T. tetragonoides* plants (Figure 6A). Overall, several *TtTLP*s, including *TtTLP6*, *TtTLP10*, *TtTLP11*, *TtTLP15*, *TtTLP16*, *TtTLP22*, and *TtTLP34*, exhibited relatively high expression patterns in all five tested organs. *TtTLP6*, *TtTLP11*, *TtTLP15*, and *TtTLP16* presented the highest expression levels in flowers, while *TtTLP10* and *TtTLP17* were highest in the young fruit of *T. tetragonoides*. Compared with vegetative organs (roots, stems, and leaves), *TtTLP*s often showed relatively higher expression levels in generative organs (flower buds and young fruit).

We also performed gene expression analysis in different tissues of *T. tetragonoides* seedlings under various stress challenges, including heat (45 °C) (Figure 6B), salt (600 mM NaCl), alkalinity (150 mM NaHCO_3_, pH 8.2), and high osmotic stress (simulated drought with 300 mM mannitol) (Figure 7), mainly based on the natural habitat of *T. tetragonoides*. Heat stress for *T. tetragonoides* plants only continued for 2 h because longer treatment (45 °C for 2 d) led to obvious wilting. Overall, short-term heat stress (45 °C for 2 h) down-regulated the expression of some *TtTLP*s in all three tissues, and only *TtTLP19* and *TtTLP35* were induced by heat stress (Figure 6B). In the *T. tetragonoides* root samples (Figure 7A), long-term alkalinity stress caused extreme suppression of the expression of all *TtTLP*s, which might be caused by the strong toxic effect of the 150 mM NaHCO_3_ liquid in which roots were submerged. Except for this, high salt stress for 2 h and 2 d, alkalinity stress for 2 h, and high osmotic stress for 2 h and 2 d also changed the expression of some *TtTLP*s in stem and leaf samples (Figure 7B,C). Of interest, the expression of *TtTLP19*, *TtTLP31*, and *TtTLP35* was induced by these challenges, suggesting the possible protective roles of these osmotin proteins in *T. tetragonoides* roots.

In addition to in silico analysis, the expression profiles of stress-responsive *TtTLPs* were validated by quantitative reverse transcription PCR (qRT-PCR) (Figure 8 and Figure 9). First, the transcripts of six *TtTLPs* were tested under a heat challenge. Overall, *TtTLP19* expression was strongly induced by heat stress. In leaf samples, the expression of *TtTLP10* and *TtTLP12* was also induced, while the expression of *TtTLP6*, *TtTLP11*, and *TtTLP22* was decreased slightly by heat stress (Figure 8).

Quantitative RT-PCR analysis was performed for six *TtTLP*s (or *TtTLP* gene pairs) using gene-specific primers at similar salt, alkalinity, and high osmotic stress treatments to validate the expression profile (Figure 9). Overall, the results agreed with the expression observed using RNA-seq data, with a few exceptions. In the case of these challenges, high alkalinity appeared to be the strongest inducing factor for the expression of specific *TtTLP*s, especially in stem and leaf samples. High salinity and osmotic stress also caused the increased expression of some *TtTLP*s; their expression was mainly focused on the aboveground parts, including stem and leaf samples.

### 2.7. Abiotic Stress Tolerance of Yeast-Heterologous-Expressing TtTLPs

We performed functional identification of several candidate *TtTLPs* using the yeast heterologous expression system. In brief, we mainly chose the candidate *TtTLPs* according to their expression patterns, and those gene members with obviously regulated expression changes due to development or stress were expected to play crucial roles in vivo. According to this principle, along with the sequence alignment of TtTLPs, six *TtTLPs*, namely *TtTLP6*, *TtTLP10*, *TtTLP11*, *TtTLP12*, *TtTLP19*, and *TtTLP22*, were cloned and inserted into pYES2. The genes *TtTLP8*, *TtTLP15*, *TtTLP16*, *TtTLP17*, and *TtTLP34* showed a high degree of homology with *TtTLP6*, *TtTLP10*, *TtTLP11*, *TtTLP12*, and *TtTLP22*, respectively, which generated degenerate primer pairs for gene cloning and the qRT-PCR assay. Here, we only showed the functional identification of the former (*TtTLP6*, *TtTLP10*, *TtTLP11*, *TtTLP12*, *TtTLP19*, and *TtTLP22*).

Plant TLPs are important proteins involved in multiple stress resistance in plants [8,10]. Therefore, we identified the salinity, alkalinity, high osmotic stress, heat, and freeze tolerance of transgenic yeast overexpressing different *TtTLPs* (Figure 10). The six expression vectors in *TtTLPs*-pYES2 and empty vector pYES2 (as control) were transformed into wild-type (WT) yeast, and the stress tolerance tests were performed with yeast spot assays under different challenges. Half of the tested *TtTLPs* (*TtTLP6*, *TtTLP12*, and *TtTLP22*) exhibited obvious sensitivity to salinity at low (0.8 M), moderate (1 M), and high (1.2 M) NaCl levels. *TtTLP10* and *TtTLP21* overexpression in yeast did not appear to impact salt tolerance, while only *TtTLP19* presented slightly elevated NaCl tolerance (Figure 10A). As for alkalinity tolerance, only *TtTLP6*, *TtTLP11*, and *TtTLP19* showed slightly elevated NaHCO_3_ tolerance, and *TtTLP10* seemed to be sensitive to NaHCO_3_ (Figure 10B). Similarly, for high osmotic stress caused by mannitol, the expression of *TtTLP6*, *TtTLP10*, *TtTLP12*, and *TtTLP22* in yeast showed more visible sensitivity than salt stress, while *TtTLP11* and *TtTLP19* did not seem to change the high osmotic tolerance of yeast, even under 1.2 M mannitol stress (Figure 10C). For the heat challenge, *TtTLP10* and *TtTLP12* caused sensitivities after 52 °C treatment (Figure 10D), and the freeze–thawing test also demonstrated that *TtTLP10* and *TtTLP12* might cause sensitivities to cold stress (Figure 10E).

Plant TLPs are secretory proteins in plant cells that are synthesized as precursors with an N-terminal signal peptide that mediates transport across the endoplasmic reticulum membrane and is then transported to other organelles or extracellularly [30,31]. Here, we hypothesized that the specific localization pattern of different TtTLPs controlling the osmotic pressure change in the different zones in yeast cells had a specific effect on stress tolerance. We also checked the H_2_O_2_ tolerance with H_2_O_2_-sensitive mutants *skn7∆* and *yap1∆* (Figure 11). Different *TtTLPs* also presented entirely different changes for H_2_O_2_ tolerance. *TtTLP6* and *TtTLP11* improved the H_2_O_2_ tolerance, while *TtTLP10* caused H_2_O_2_ sensitivity in yeast (Figure 11A). In *yap1∆* yeast, *TtTLP10* and *TtTLP12* hardly influenced the H_2_O_2_ tolerance, while *TtTLP6*, *TtTLP11*, *TtTLP19*, and *TtTLP22* slightly improved it (Figure 11B).

Plant *TLP*s have also been speculated to play important roles in heavy metal (HM) detoxification since they encode cysteine-rich proteins [32], and some research has demonstrated that *TLP* expression is induced by HMs, including cadmium [32] and lead [33]. Here, we also checked the metal detoxification abilities of *TtTLPs* with yeast heterologous expression systems. Expression vectors *TtTLPs*-pYES2 and pYES2 were transformed into different yeast strains, including WT and several mutants (*ycf1∆*, *zrc1∆cot1∆*, *cot1∆*, *smf1∆*, and *pmr1∆*). As shown in Figure 12A, only *TtTLP10* presented slight increases in the Cd tolerance of the Cd-sensitive mutant strain *ycf1∆* on SDG plates with a low Cd concentration (30 μM), and under moderate (40 μM) or high (50 μM) Cd concentrations, none of the *TtTLPs* improved the Cd tolerance of this yeast mutant strain. The other five *TtTLPs*, namely *TtTLP6*, *TtTLP11*, *TtTLP12*, *TtTLP19*, and *TtTLP22*, significantly increased the Cd sensitivity of *ycf1∆*. Zn tolerance was also detected in double mutant *zrc1∆cot1∆*, and only *TtTLP22* slightly increased the Zn tolerance under a moderate concentration (0.3 mM). Under low Zn challenge (0.2 mM), *TtTLP6*, *TtTLP10*, *TtTLP19*, and *TtTLP22* all caused Zn sensitivity in *zrc1∆cot1∆* (Figure 12B). Similarly, *TtTLP6*, *TtTLP11*, and *TtTLP22* generated a slight Co sensitivity in *cot1∆* (Figure 12C). Under the Ni challenge, our results showed that only *TtTLP22* elevated Ni sensitivity at high concentrations (1 mM). Under low (0.25 mM) or moderate (0.5 mM) Ni challenges, none of the *TtTLPs* changed the Ni tolerance or sensitivity (Figure 12D). For Mn tolerance, the different *TtTLPs* presented diverse phenotypes. *TtTLP6*, *TtTLP10*, and *TtTLP12* significantly increased the Mn tolerance of yeast, while *TtTLP11*, *TtTLP19*, and *TtTLP22* did not affect Mn tolerance (Figure 12E). Metal tolerance mediated by *TtTLPs* had member-specific characteristics. Given the limitations of the yeast expression system, the relative conclusions in plant cells need further confirmation through transgenic assays in plants.

## 3. Discussion

As an excellent and popular medicinal and edible vegetable resource, *T. tetragonoides* can also be used for the restoration of the ecological environment in some tropical coastal regions and coral islands in the landscape [34,35]. Due to the harsh natural habitats of *T. tetragonoides* plants, this species can be planted in tropical coastal areas, replacing traditional crops and vegetables, and helping meet the basic needs of people, thereby becoming a important plant resource with irreplaceable economic values. To cope with these unfavorable environmental conditions, *T. tetragonoides* adopted various resistance strategies using morphological and physiological mechanisms. For example, *T. tetragonoides* is an inward secretohalophyte and can store excess salinity in the salt gland or salt bladder, which is widely distributed in the epidermal cells of *T. tetragonoides* leaves and stems. This salt isolation strategy is also an efficient mechanism for dealing with hypersaline and hypertonic environments [36,37]. Coupled with the specialization of vegetative morphology, the intrinsic molecular mechanism of *T. tetragonoides* plants for habitat adaptation impacts the genome, especially functional genes. Therefore, it is quite necessary to understand the mechanisms by which *T. tetragonoides* has adapted to extreme environmental abiotic stress, including drought, high salinity–alkalinity, and high temperature stress, to complete its life cycle.

Plant TLPs, also called osmotins, belong to the cysteine-rich proteins (CRPs). Similar to other plant CRPs, such as thionins and lipid transfer proteins (LTPs) [38], TLPs are classified as pathogenesis-related protein family 5 (PR-5) due to their induced expression and are involved in defense systems against various biotic stressors [39]. The main characteristic of TLPs is their thaumatin_2 domains with 16 conserved cysteine (Cys) residues, which form eight intra-molecular disulfide bonds and are involved in providing stability to the proteins under extreme pH and temperature ranges [20]. Previous studies have indicated that plant TLPs are also involved in the regulation of abiotic stress, such as drought [15,40], salt [10,17], and temperature challenges [17,18,41]. Here, considering the native habitats of *T. tetragonoides* causing constant osmotic pressure on this species, we selected the *TtTLP* family to research the ecological adaptability of *T. tetragonoides*, mainly concerning extreme drought and high salinity/alkalinity.

Although there have been some relevant studies on the *TLP* gene families in many plants [8,18,19,20,21,22,23,24,25,26], further research on the specific functions of these genes is still needed, especially in special habitat plants, such as halophytes or other special habitat plant species [42,43]. Related transgenic research on the *TLP*s of wild plant origins has provided valuable information to improve abiotic stress tolerance in rice [14], tobacco [18], Arabidopsis [7,19], and other plants [13]. In this study, 37 *TtTLP* genes were identified from the whole-genome sequence of *T. tetragonoides*. The genomic DNA and deduced protein sequences of *TtTLP*s were compared with each other and with their homologous genes in Arabidopsis and rice, establishing inter-genomic and intragenic phylogenetic trees (Figure 2 and Figure 4). Detailed analyses were also performed to identify gene chromosomal locations (Figure 1), gene structures (Figure 3), and *cis*-acting elements in promoter regions (Figure 5). Primary sequences, physiochemical properties, subcellular localization, transmembrane domains, and motifs of the proteins were also summarized (Table 1). Our results provide insight into the *TtTLP* genes involved in plant abiotic stress processes and provide genetic resources for further transgenic improvement in crops and other economic plants.

Gene duplication is considered a main driving force of genome evolution, and segmental and tandem duplication are regarded as two main driving forces for gene family expansion in plants. In this study, duplication event analyses were carried out to understand their roles in the evolution and expansion of the *TtTLP* gene family (Figure 2 and Figure 4, Appendix A). According to the phylogenetic tree, most of the *TtTLP* members existed in pairs, except *TtTLP18*, *TtTLP19*, *TtTLP24*, *TtTLP31*, and *TtTLP35*, which were distinctly different from those in Arabidopsis and rice *TLP* families (Figure 4). This further suggests that duplications of *TtTLP*s are a major factor contributing to species distributional specificity.

The biological functions of genes are mainly shown as regulatory expression patterns under different promoters, which can be bound directly or indirectly by specific regulatory factors, mainly TFs, such as WRKY, MYB, MYC, HSF, and NAC. The CEs existing in gene promoter regions play crucial roles in controlling their expression patterns. The *TtTLP* promoter sequences were systematically analyzed, and specific CEs were summarized (Figure 5). The abiotic stress-related CEs extensively existed in the *TtTLP* promoters, including anaerobic-responsive elements, as-1 (oxidative stress-responsive), HSE (heat-responsive), LTRE (low temperature-responsive), MYB (MYB-binding site), and MYC (MYC-binding site). Biotic stress-related CEs (TC-rich repeat, MeJA-responsive element, salicylic acid-responsive element) are also relatively common. Other hormone-related CEs, such as gibberellin-responsive elements, auxin-responsive elements, ABRE (ABA-responsive elements), and ERE (ethylene-responsive elements), were also widespread. This could also be considered an adaptation mechanism to stress conditions in *T. tetragonoides* in vivo, mediated by the *TtTLP* family.

Low-molecular-weight TLPs have been identified in different plants [18,19,20,21,22,23,24,25,26], and many functional studies have shown their crucial roles in regulating plant growth, development, and defense against pathogens [8,9]. However, the roles and detailed regulation mechanisms of the plant *TLP* family under abiotic stress remain unclear. The heterogeneous overexpression of plant *TLP*s has been performed in many species [7,12,13,14,15,16,17,18], resulting in elevated tolerance to salt, drought, and other challenges and even sensitivities to abiotic stressors [19]. In the present study, we adopted a galactose-induced yeast system to verify the tolerance caused by the expression of specific *TtTLP*s, which is convenient for gene functional identification in vivo in the short term. In single-cell yeast, tolerance to stress, including salt, alkalinity, high osmotic pressure, heat, freezing, and oxidative stress, was significantly reduced or slightly improved by *TtTLP*s (Figure 10 and Figure 11). We suspect that this may be related to the subcellular localization of specific TtTLPs in yeast. If these TtTLPs all belonged to secretory proteins and most of them were located around the cell, the changes in osmotic pressures caused by the surrounding TtTLPs would affect the relative growth activities of yeast cells by changing the hydraulic potential, thereby inhibiting the growth of yeast cells. This is quite different from the phenotype of plant-overexpressing *TLP*s, most of which showed elevated tolerance to high salinity or drought [12,13,14,15,16,17]. This can be attributed to the fact that plants are multicellular organisms, and the overexpression of specific *TLP*s causes the wide distribution of TLPs in vivo, thereby changing the abilities of the whole plant to respond to stress challenges.

Plant TLPs are cysteine-rich proteins, and they might be involved in HM detoxification [32]. Here, we checked the HM tolerance of yeast caused by the expression of specific *TtTLP*s (Figure 12), and the tolerance to specific metals was member-specific. This may occur because of the specific subcellular localization of TtTLPs in yeast. Further combining the features of different yeast mutant strains, including the deletion of specific metal chelation or transportation proteins, such as yeast ycf1, which is an ATP binding cassette (ABC) transporter [44] and yeast pmr1, a Golgi-localized Mn^2+^ transporter [45], showed specificity in the tolerance to different metals in different mutant strains.

*Tetragonia tetragonioides* is native to tropical and subtropical coral regions and shows strong adaptability to high salinity/alkalinity, seasonal drought, heat, and other extreme circumstances. Our study presents possible genetic resources for stress tolerance in this species, the *TtTLP* family. The related plant transgenic assay will further demonstrate the detailed functions of *TtTLP*s to be used as candidate genes for crop genetic improvement. These findings suggest that *TtTLP*s undertake abiotic stress tolerance biological functions and play pivotal roles for the ecological adaptability of *T. tetragonoides*.

## 4. Materials and Methods

### 4.1. Plant Materials and Stress Treatments

*Tetragonia tetragonoides* seeds and plants were collected from the coastal areas of Guangdong and Fujian provinces in China and then cultivated in the South China National Botanical Garden (SCNBG) by Shuguang Jian (JS) and Mei Zhang (ZM). *Tetragonia tetragonoides* seeds were germinated in wet vermiculite, and seedlings were grown at 26 °C under a 16/8 h regular dark/light cycle with 50–60% humidity. The tissue-specific transcriptional patterns of *TtTLP*s were analyzed with roots, stems, leaves, flowers, and fruit organs gathered from adult *T. tetragonoides* plants. The expression patterns of *TtTLP*s under various stress challenges were further detected with roots, stems, and leaves collected from 60 d-old *T. tetragonoides* seedlings. For different stress challenges for seedlings, the young *T. tetragonoides* plants were exposed to different stress conditions, including heat (45 °C), high osmotic stress (300 mM mannitol), high salt stress (600 mM NaCl), and high alkalinity stress (150 mM NaHCO_3_, pH 8.2). In brief, for the heat treatment, *T. tetragonoides* seedlings were moved to a 45 °C illumination incubator for 2 h. For the other three stress challenges, the *T. tetragonoides* seedlings were removed from their vermiculite pots, carefully washed with water to remove matrix from the roots, and transferred to challenge solutions with the roots being submerged. The roots, stems, and young leaves from the *T. tetragonoides* seedling were collected 2 and 48 h after stress treatments, with the unchallenged *T. tetragonoides* seedlings’ tissues (0) used as a control. All samples were immediately frozen in liquid nitrogen after picking and stored at −80 °C for further experiments. Three independent biological replicates were used. The RNA sequencing (RNA-seq) data from *T. tetragonoides* used for in silico expression analyses were generated under the same conditions.

### 4.2. Identification and Characterization of TtTLPs in T. tetragonoides

The *T. tetragonoides* genome was sequenced and submitted to the NCBI database (NCBI accession number: JBBMRK000000000, unreleased). All *T. tetragonoides* proteins were identified with InterProscan (https://www.ebi.ac.uk/interpro/search/sequence/, accessed on 20 November 2023), and the conserved domains and motifs (e < 1 × 10^−5^) were assessed. To characterize the TtTLP family members, the conserved TLP domain (pfam No. PF00314, PROSITE profile No. PS51367, or InterPro No. IPR001938) was searched as a model, and the protein sequences containing this domain were screened using HMM3.0 software. The domains were also confirmed using the NCBI CDD program (https://www.ncbi.nlm.nih.gov/cdd/, accessed on 20 November 2023). The *T. tetragonoides* proteins with the TLP domain and corresponding genes were eventually identified as the TtTLP family.

### 4.3. Multiple Sequence Alignment and Phylogenetic Analysis of TLP Proteins

Multiple sequence alignments of candidate plant TLPs, including the 37 TtTLPs from *T. tetragonoides*, 27 AtTLPs from *Arabidopsis thaliana*, and 31 OsTLPs from rice (*Oryza sativa*) [23], were performed with 1000 bootstrap replicates using MEGA X software (Version 10.2, https://www.megasoftware.net/, accessed on 20 November 2023) with ClustalW. The *A*. *thaliana* TLP sequences were obtained from the Arabidopsis Information Resource (TAIR, http://www.arabidopsis.org, accessed on 20 November 2023), and the *O*. *sativa* TLPs were obtained from the Rice Genome Annotation Project (RGAP, http://rice.plantbiology.msu.edu/index.shtml, accessed on 20 November 2023) database. The phylogenetic tree was constructed using the neighbor-joining (NJ) method with TtTLPs, AtTLPs, and OsTLPs. The obtained TLP nucleotide and protein sequences from these three species are listed in Appendix A.

### 4.4. Bioinformatic Analysis of T. tetragonoides TLP Genes

The *TtTLP* genes’ exon–intron structures were determined based on alignments of the coding regions and full-length sequences with the online program Gene Structure Display Server 2.0 (https://gsds.gao-lab.org/Gsds_about.php, accessed on 20 November 2023). The bioinformatics tools WoLF_PSORT (https://www.genscript.com/wolf-psort.html, accessed on 20 November 2023) and Plant-mPLoc (http://www.csbio.sjtu.edu.cn/bioinf/plant-multi, accessed on 20 November 2023) were used to predict the subcellular localization of each TtTLP. The TtTLP motifs were predicted using MEME (http://meme-suite.org/index.html, accessed on 20 November 2023), with a maximum number of motifs of 10 and an optimum motif width of 20–50 residues. The identified TtTLP sequences were used to calculate the molecular weight (MW) and isoelectric point (pI) of the proteins using Expasy (https://web.expasy.org/protparam/, accessed on 20 November 2023). The 3D structures of the TtTLPs were analyzed using the Phyre^2^ program (http://www.sbg.bio.ic.ac.uk/phyre2/html/page.cgi?id=index, accessed on 20 November 2023).

### 4.5. Promoter Sequence Profiling of TtTLPs

Putative *TtTLP* promoter regions (2000 bp upstream of ATG) were retrieved from the *T. tetragonoides* genome database (Appendix A) and uploaded to the PlantCARE database (http://bioinformatics.psb.ugent.be/webtools/plantcare/html/, accessed on 1 March 2024) for *cis*-acting element (CE) analysis. The CEs were classified as either hormone-specific (gibberellin-responsive elements, MeJA-responsive elements, auxin-responsive elements, salicylic acid-responsive elements, EREs, and ABREs) or abiotic stress-responsive (light responsive elements, MYCs, MYBs, MBSs, TC-rich repeats, and LTREs). The CEs are summarized in Appendix A. Several selected *TtTLP* promoters were visualized using TBtools [46].

### 4.6. Expression Profiling of TtTLPs in Different Organs or Under Specific Challenges

A transcriptome database was constructed for *T. tetragonoides* using Illumina HiSeq X sequencing technology. The quality of the RNA-seq datasets created from five tissues (roots, stems, young leaves, flowers, and young seeds collected from *T. tetragonoides* growing in the SCNBG) was examined using FastQC (http://www.bioinformatics.babraham.ac.uk/projects/fastqc/, accessed on 1 March 2024), which produced 40 Gb of clean reads. Clean reads were mapped to the *T. tetragonoides* reference genome using Tophat v.2.0.10 (http://tophat.cbcb.umd.edu/, accessed on 1 March 2024). The fragments per kilobase of transcript per million mapped reads (FPKM) values were used to calculate the gene expression levels according to the length of the gene and the read counts mapped to the gene: FPKM = total exon fragments/[mapped reads (millions) × exon length (kb)]. The expression levels [log2 (FPKM + 1)] of TtTLPs were visualized as heatmaps using TBtools. The FPKM values for all samples are listed in Appendix A.

The qRT-PCR was also performed to detect the transcript abundance of several TtTLP transcripts. In brief, the total RNAs were isolated from different *T. tetragonoides* seedling tissues after specific stress treatments and reverse transcribed to cDNA; the untreated *T. tetragonoides* seedling tissues were used as controls. The total RNA samples were extracted using an EasyPure^®^ Plant RNA Kit (TransGen Biotech, Beijing, China). The RNAs were quantified using a NanoDrop1000 (NanoDrop Technologies, Inc., Wilmington, DE, USA) spectrophotometer, and their integrity was checked on a 0.8% agarose gel. After that, the cDNAs were synthesized using the cDNA Synthesis SuperMix kit (TransGen Biotech, Beijing, China) following the manufacturer’s instructions. Quantitative RT-PCR was conducted with the LightCycler480 system (Roche, Basel, Switzerland) and the TransStart Tip Green qPCR SuperMix (TransGen Biotech, Beijing, China). All TtTLP genes’ expression data obtained via qRT-PCR were normalized to the reference gene TtACT’s expression (NCBI accession No.: MH33308). The primers used for qRT-PCR (TtACTRTF/TtACTRTR as the reference gene and other TtTLP-specific primer pairs) were listed in Appendix A.

### 4.7. In Vivo Stress Tolerance Assay for TtTLP Overexpression in Yeast

Several *TtTLP*s were PCR cloned with a cDNA sample of *T. tetragonoides* as a template. In brief, the open reading frames (ORFs) of candidate *TtTLP*s were PCR amplified with gene-specific primer pairs (Appendix A). The PCR fragments were purified and then inserted into the *Bam*HI and *Eco*RI sites of the yeast expression vector pYES2 to yield recombinant plasmids of TtTLPs-pYES2 and sequenced. The different yeast strains were used in this study, including wild-type (WT) *Saccharomyces cerevisiae* (BY47471; MATa; his3Δ1; leu2Δ0; met15Δ0; ura3Δ0), heavy metal (HM)-sensitive mutant strains cot1Δ, smf1Δ, ycf1Δ, pmr1Δ, double-mutant strain *zrc1Δ/cot1Δ* (BY4741; *MATa*; *his3∆1*; *leu2∆0*; *met15∆0; ura3∆0*; *zrc1::natMX*; *cot1::kanMX4*), and H_2_O_2_-sensitive mutant strains *skn7Δ* and *yap1Δ*. The WT (Y00000) and *cot1Δ* (Y01613), *smf1Δ* (Y06272), *ycf1Δ* (Y04069), *pmr1Δ* (Y04534), *skn7Δ* (Y02900), and *yap1Δ* (Y00549) were all obtained from Euroscarf (http://www.euroscarf.de/index.php?name=News, accessed on 1 June 2022). The double-mutant strain zrc1Δ/cot1Δ was obtained from Yuan’s lab [47]. The standard polyethylene glycol (PEG)-lithium acetate-based transformation procedure was used for yeast plasmid transformation with amino acid defect screening. The yeast spot assays for NaCl, NaHCO_3_, mannitol, and HM tolerance were performed as previously described [48]. For the heat or freeze tolerance tests of yeast strains, the yeast cultures with different specific OD_600_ values were placed on a thermostat (52 °C) for 20, 30, and 40 min for heat challenges. The yeast cultures were quickly frozen in liquid nitrogen and slowly thawed at room temperature, and this operation was repeated one, two, or three times to test the freezing tolerance.

### 4.8. Statistical Analysis

All experiments in this study were repeated independently three times, and the results are shown as the mean ± standard deviation (SD) (*n* ≥ 3). Pairwise differences between means were analyzed using a Student’s *t*-test in Microsoft Excel 2010.

## 5. Conclusions

In the present study, 37 *TtTLP* genes were identified in the *T. tetragonoides* genome. The *TtTLP* family members were responsive to multiple stressors. Several *TtTLP*s were cloned and functionally identified using the yeast heterogeneous expression system, followed by testing for salt tolerance and osmotic adjustment in cells, as well as detoxification of several heavy metals. Overall, the results highlighted the roles of *TtTLP*s in the ecological adaptability of *T. tetragonoides* to tropical coastal regions. Our results also suggested *TtTLP*s as useful candidates for providing multiple stress tolerances in crops.

## Figures and Tables

**Figure 1 plants-13-02355-f001:**
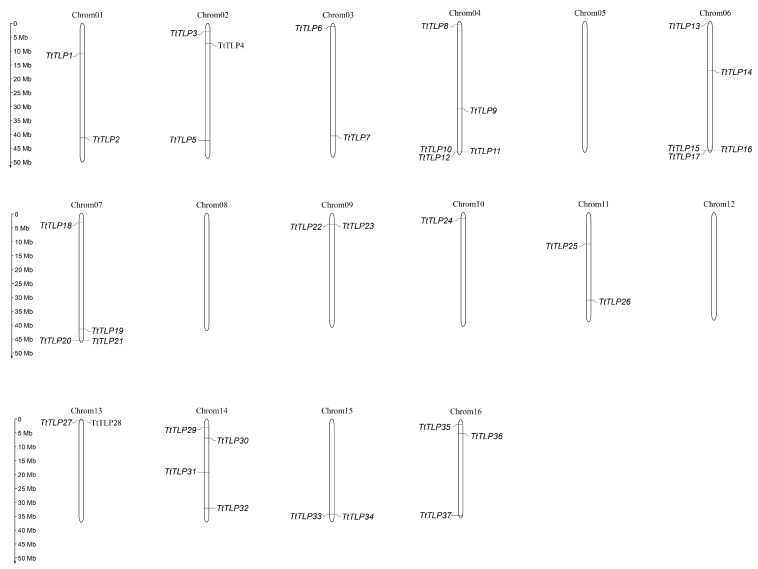
Locations of the 37 *TtTLP*s on 16 chromosomes in *T. tetragonoides*.

**Figure 2 plants-13-02355-f002:**
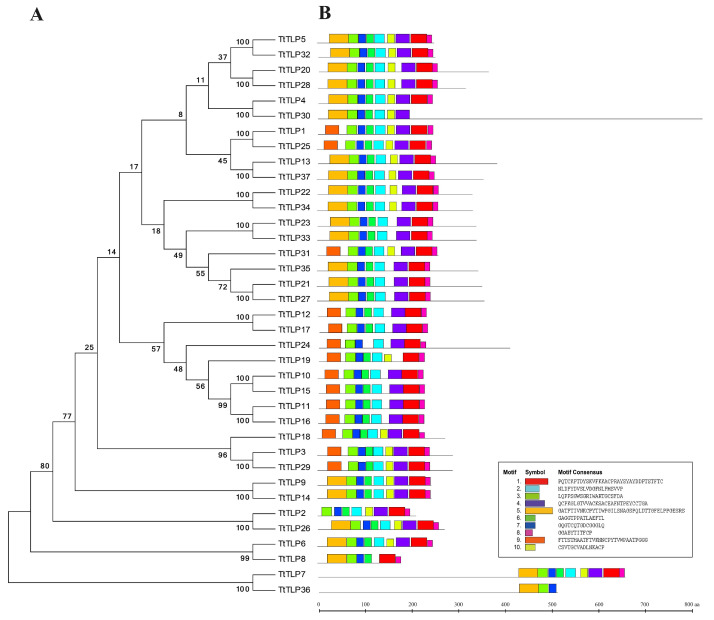
(**A**) Phylogenetic relationships of the 37 TtTLPs from *T. tetragonoides*. (**B**) The conserved motifs of each group of TtTLPs identified using the MEME web server. Different motifs are represented by different colored boxes, and the motif sequences are provided at the bottom.

**Figure 3 plants-13-02355-f003:**
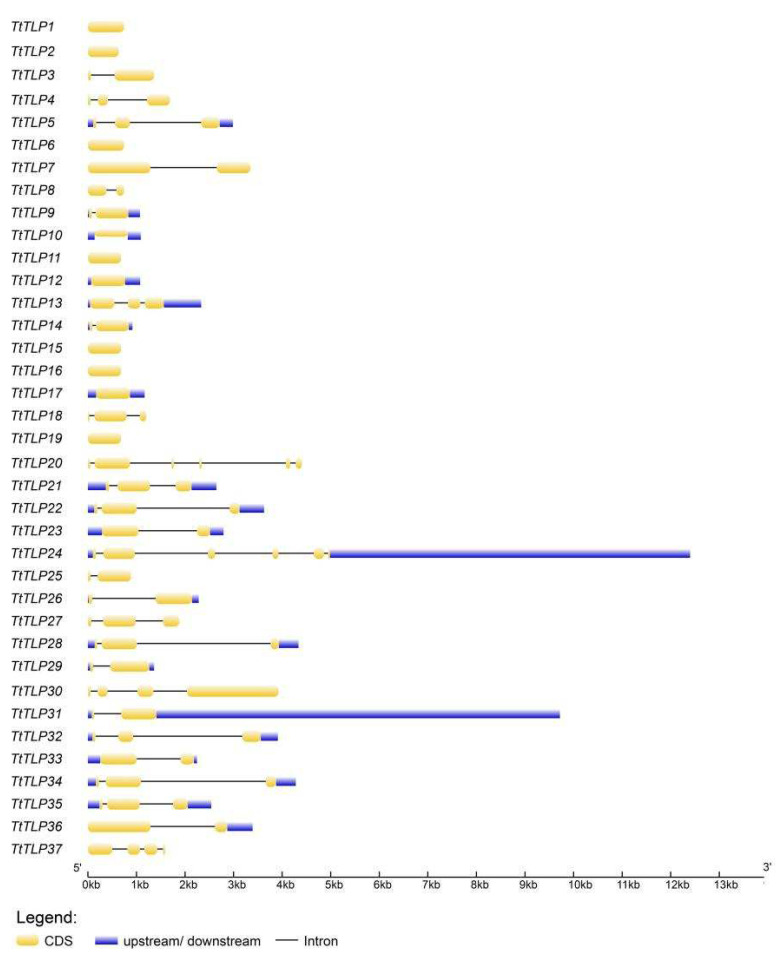
Exon–intron organization of the *TtTLP* genes constructed using GSDS 2.0.

**Figure 4 plants-13-02355-f004:**
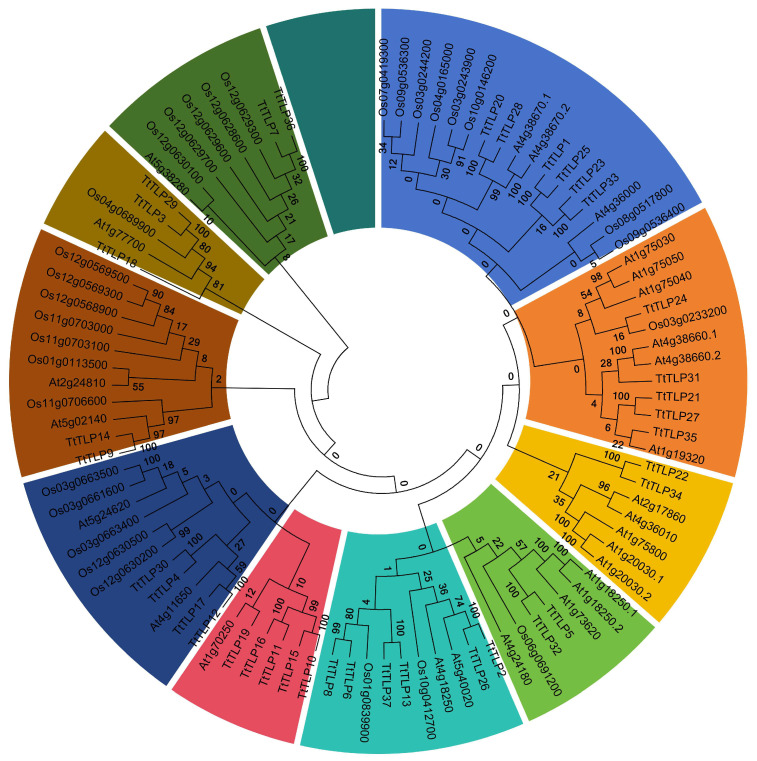
Phylogenetic relationships of the 37 TtTLPs from *T. tetragonoides*, 27 AtTLPs from *Arabidopsis thaliana*, and 30 OsTLPs from *Oryza sativa*. The amino acid sequences of these 94 TLPs from three plant species were compared with ClustalW alignment, and the phylogenetic tree was constructed in MEGA X using the neighbor-joining method, with 1000 bootstrap repetitions. The different branch colors represent different subgroups.

**Figure 5 plants-13-02355-f005:**
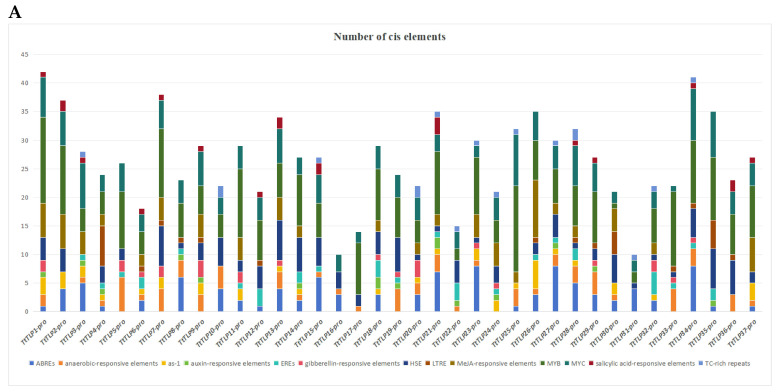
Statistics for predicted *cis*-acting elements in the *TtTLP* promoters (ATG_upstream 2000 bp). (**A**) Summaries of the 13 *cis*-acting elements in the promoter regions of 37 *TtTLP*s. The scale bar represents 200 bp. (**B**) Distribution of the 13 *cis*-acting elements in the 37 *TtTLP* promoter regions. The elements are represented by different symbols. Information on these elements is listed in Appendix A.

**Figure 6 plants-13-02355-f006:**
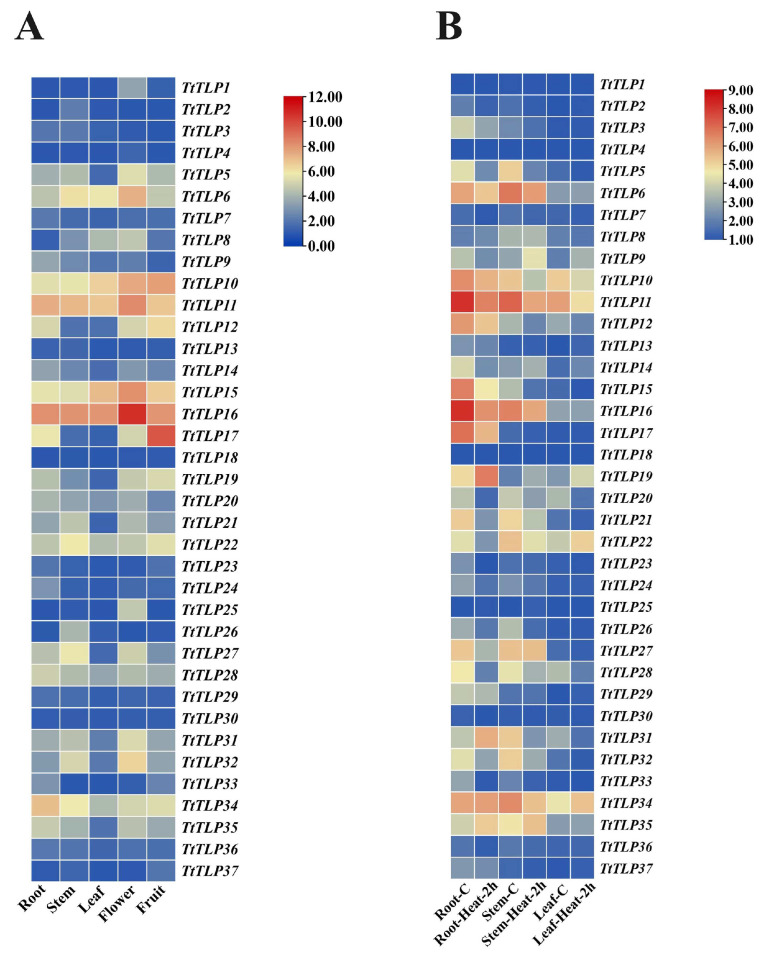
(**A**) Heatmaps showing the expression levels of the *TtTLP*s in the roots, stems, leaves, flower buds, and young fruit of *T. tetragonoides* plants. (**B**) Heatmaps showing the expression levels of the *TtTLP*s in *T. tetragonoides* seedlings under heat treatment (45 °C for 2 h). “-C” represents “control”. The RNA-seq data of the *TtTLP*s were listed in Appendix A. The heat map was constructed from log2-transformed FPKM (+1) values, and normalized treatments were carried out based on rows. The RNA-seq data of the *TtTLP*s are listed in Appendix A.

**Figure 7 plants-13-02355-f007:**
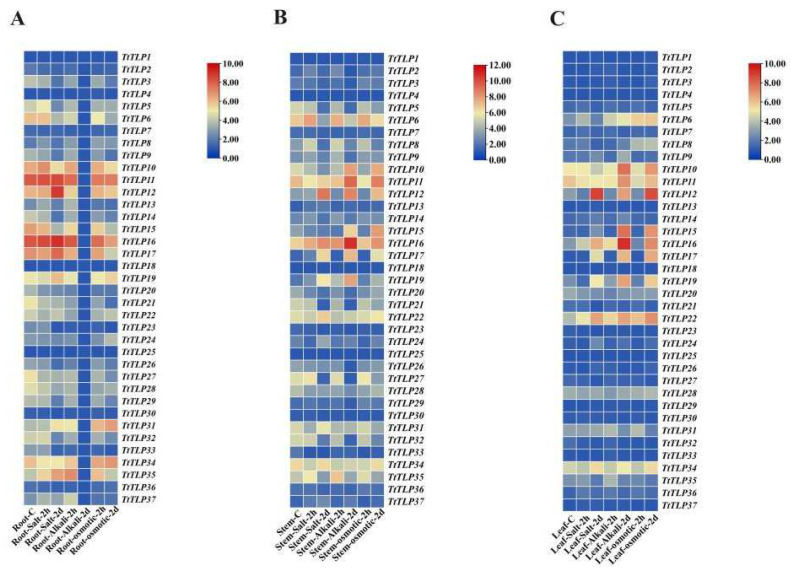
Heatmaps showing the expression levels of the *TtTLP*s under salt, alkalinity, and high osmotic treatment in *T. tetragonoides* seedling roots (**A**), stems (**B**), and leaves (**C**). “-C” “-2 h”, and “-2 d” each represent “control”, “stress treatment for two hours”, and “stress treatment for two days (48 h)”.The heat map was constructed from log2-transformed FPKM (+1) values, and normalized treatments were carried out based on rows. The RNA-seq data of the *TtTLP*s were listed in Appendix A.

**Figure 8 plants-13-02355-f008:**
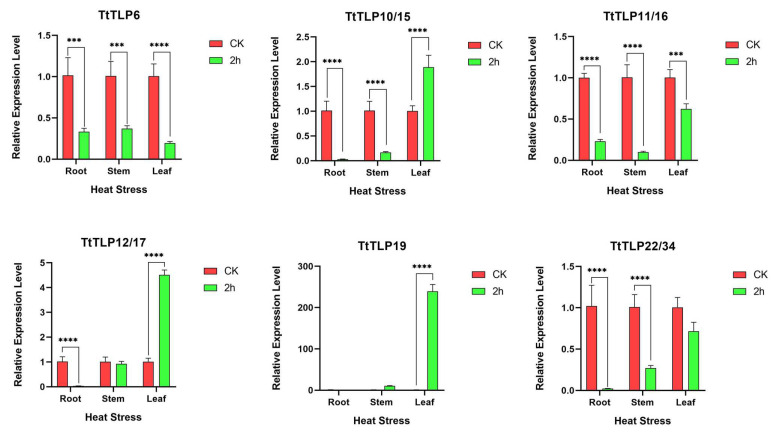
Quantitative RT-PCR detection of the expression levels of the six candidate *TtTLP*s in response to heat stress in *T. tetragonoides* seedlings. The relative expression levels in the root, stem, and leaf samples under heat stress treatment (45 °C; 0 and 2 h) were calculated using the 2^−ΔCt^ method, with the housekeeping gene *TtACT* as a reference gene. Bars show the mean values ± SD of *n* = 3–4 technical replicates. Asterisks indicate significant differences from the CK (control check, without heat stress, Student’s *t*-test, *** *p* < 0.001, and **** *p* < 0.0001).

**Figure 9 plants-13-02355-f009:**
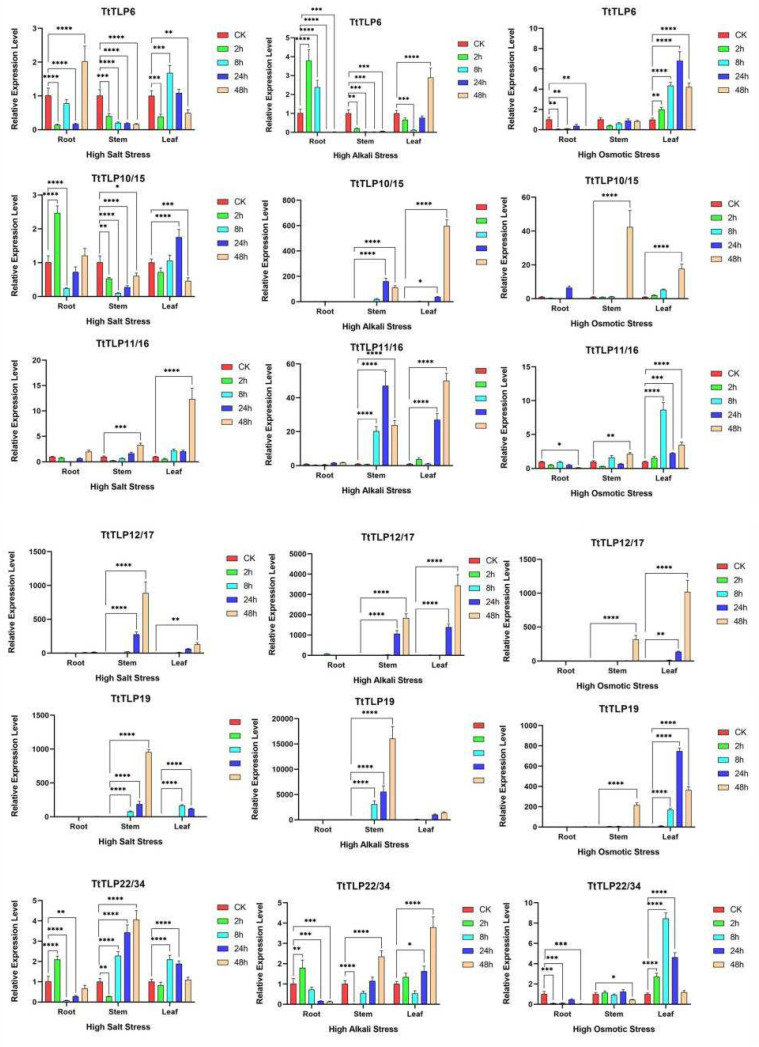
Relative expression levels of six candidate *TtTLP*s checked by qRT-PCR in root, stem, and leaf samples of *T. tetragonoides* seedlings under high salinity, high alkalinity, and high osmotic treatments (0, 2, 8, 24, and 48 h). Relative expression values were calculated using the 2^−ΔCt^ method, with the housekeeping gene *TtACT* as a reference gene. Bars show the mean values ± SD of *n* = 3–4 technical replicates. Asterisks indicate significant differences from the CK (control check, without abiotic stresses, Student’s *t*-test, * *p* < 0.05, ** *p* < 0.01, *** *p* < 0.001, and **** *p* < 0.0001).

**Figure 10 plants-13-02355-f010:**
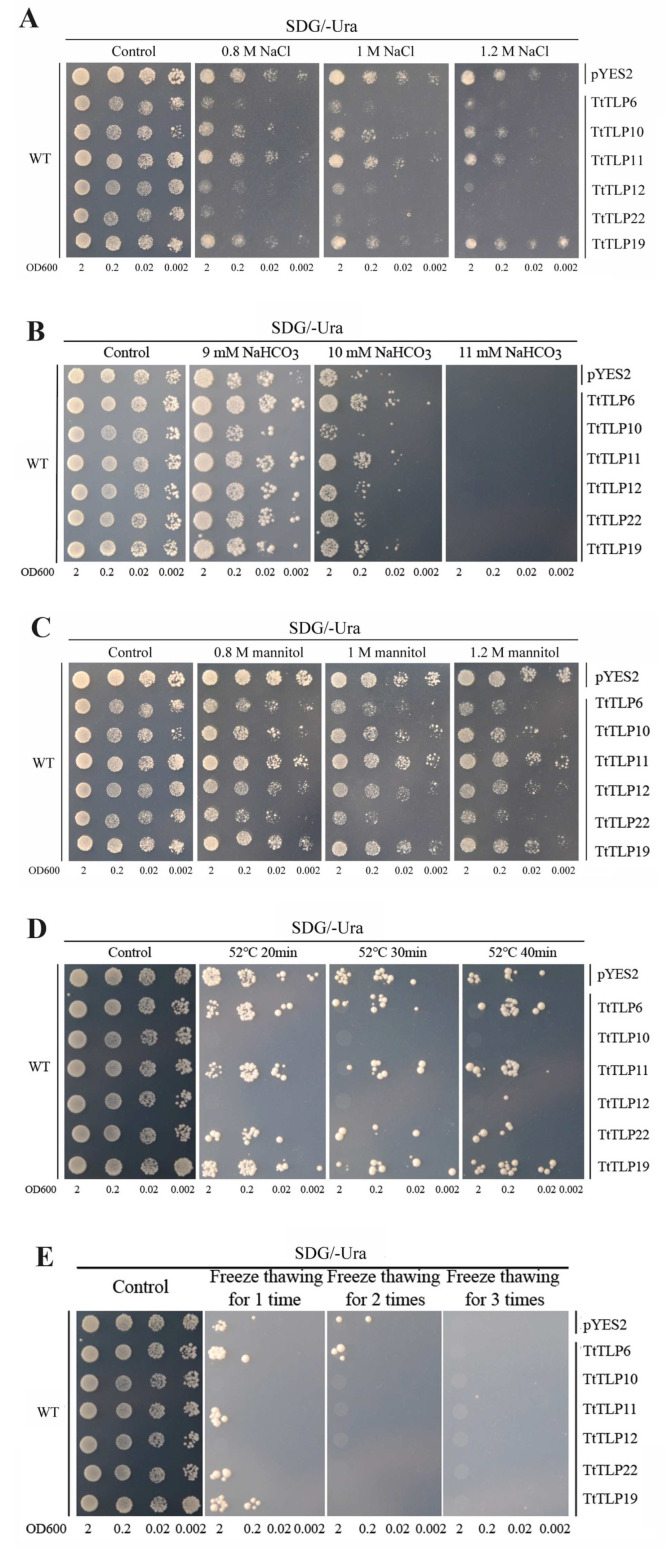
Functional identification related to abiotic stress of six candidate *TtTLP*s in yeast using a heterologous expression assay. The yeast wild-type (WT) strain BY4741 was transformed with the empty vector pYES2 or six recombinant vectors, namely TtTLP6-pYES2, TtTLP10-pYES2, TtTLP11-pYES2, TtTLP12-pYES2, TtTLP22-pYES2, and TtTLP19-pYES2. Yeast cultures were adjusted to OD600 = 2, and 2 μL serial dilutions (10-fold, from left to right in each panel) were spotted on SDG-Ura medium plates supplemented with different NaCl concentrations (0, 0.8, 1, and 1.2 M) (**A**); NaHCO_3_ concentrations (0, 9, 10, and 11 mM, pH 8.2) (**B**); mannitol concentrations (0, 0.8, 1, and 1.2 M) (**C**); heat challenges (52 °C for 20, 30, and 40 min, yeast strains without heat stress as control) (**D**); and freeze–thawing challenges (one, two, and three times, yeast strains without freeze-thawing as control) (**E**). The WT strain transformed with pYES2 was used as a positive control, and the yeast spots growing on the SDG-Ura medium plate without any challenge were the CK (check) control. The plates were incubated for 2–5 d at 30 °C.

**Figure 11 plants-13-02355-f011:**
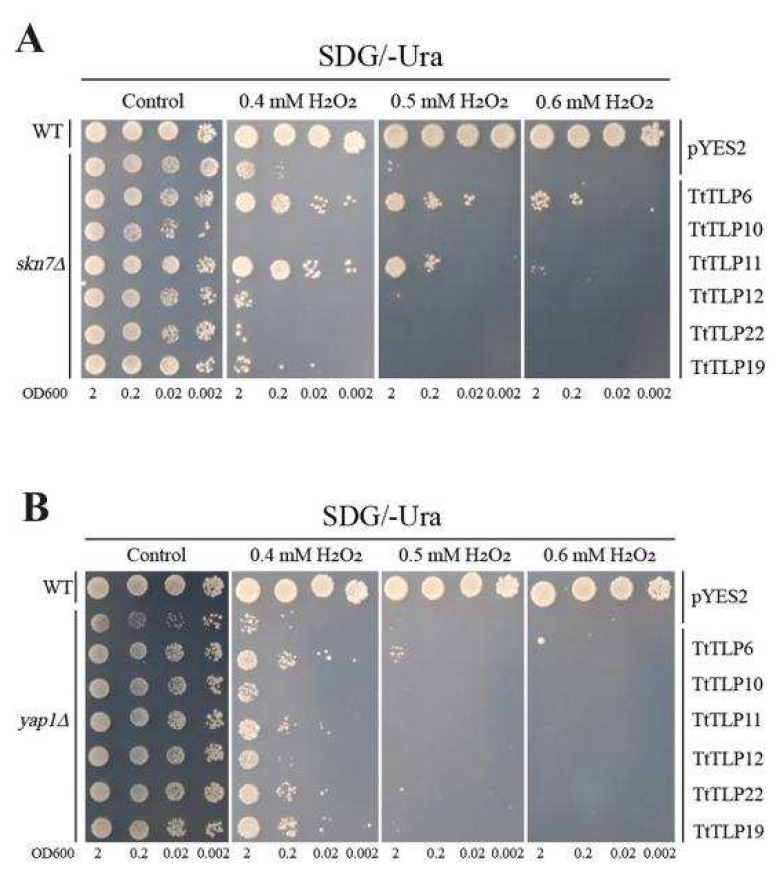
Functional identification related to hydrogen peroxide (H_2_O_2_) tolerance of six *TtTLP*s in yeast using heterologous expression assays. The yeast wild-type (WT) and two H_2_O_2_-sensitive mutant strains, *skn7∆* and *yap1∆*, were transformed with the empty vector pYES2 or six recombinant vectors, namely TtTLP6-pYES2, TtTLP10-pYES2, TtTLP11-pYES2, TtTLP12-pYES2, TtTLP22-pYES2, and TtTLP19-pYES2. (**A**) H_2_O_2_ (0, 0.4, 0.5, and 0.6 mM) tolerance in *skn7∆*; (**B**) H_2_O_2_ (0, 0.4, 0.5, and 0.6 mM) tolerance in *yap1∆*. The yeast strains were cultured and spotted, as described above.

**Figure 12 plants-13-02355-f012:**
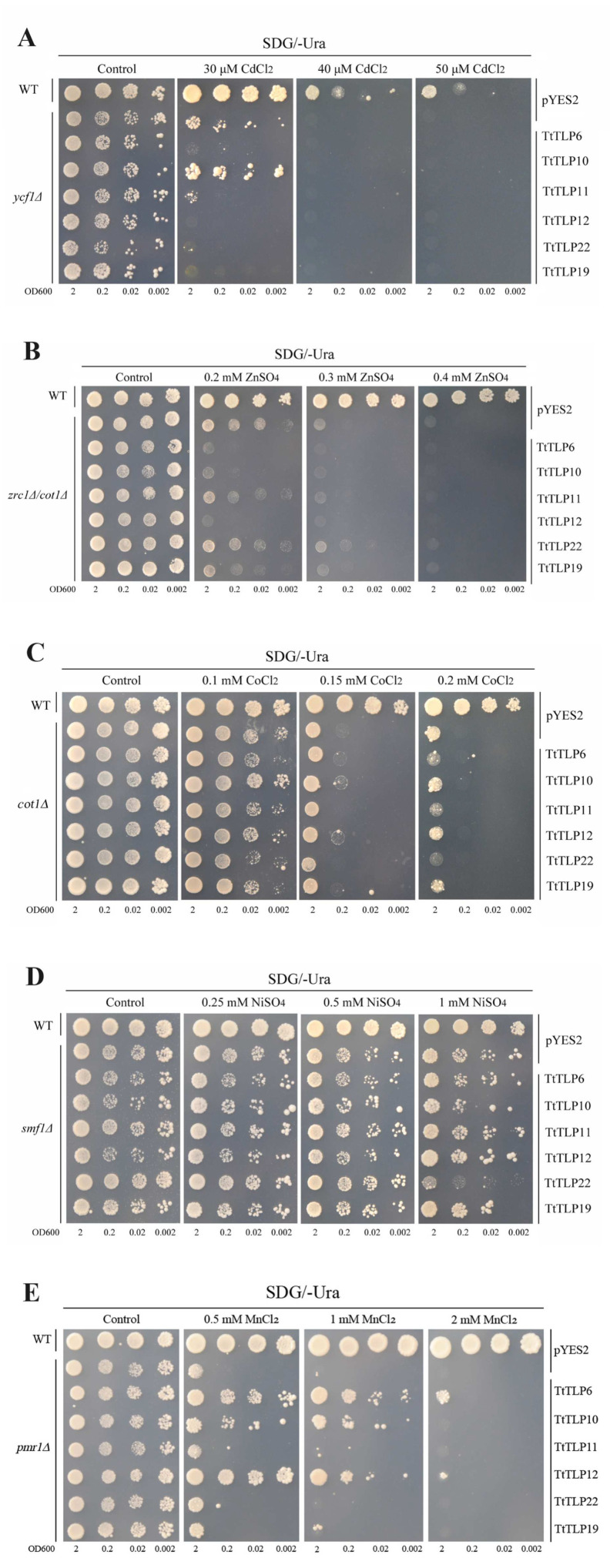
Functional identification related to heavy metal (HM) tolerance of six *TtTLP*s in yeast using heterologous expression assays. The yeast wild-type (WT) and a series of metal-sensitive mutant strains were transformed with the empty vector pYES2 or six recombinant vectors, namely TtTLP6-pYES2, TtTLP10-pYES2, TtTLP11-pYES2, TtTLP12-pYES2, TtTLP22-pYES2, and TtTLP19-pYES2. (**A**) Cadmium (Cd, 0, 30, 40, and 50 μM) tolerance in *ycf1∆*; (**B**) zinc (Zn, 0, 0.2, 0.3, and 0.4 mM) tolerance in *zrc1∆cot1∆*; (**C**) cobalt (Co, 0, 0.1, 0.15, and 0.2 mM) tolerance in *cot1∆*; (**D**) nickel (Ni, 0, 0.25, 0.5, and 1 mM) tolerance in *smf1∆*; and (**E**) manganese (Mn, 0, 0.5, 1, and 2 mM) tolerance in *smf1∆*. The yeast strains were cultured and spotted, as described previously.

**Table 1 plants-13-02355-t001:** Nomenclature, properties, and prediction of subcellular localization for TtTLPs identified from *Tetragonia tetragonoides*.

Name	Locus	Length (aa) and MW (kDa)	Major Amino Acid (%)	PI	II	AI	GRAVY	Disordered aa (%)	TMHs and Topologies	WoLF_PSORT	Plant-PLoc
TtTLP1	01G0009210	247–26.38	T (11.7%), A (8.1%), P (8.1%)	7.39	40.68	72.27	0.005	70.45	None/outside	chlo: 14	Extracellular
TtTLP2	01G0018050	211–22.82	S (10.0%), L (9.0%), V (7.6%)	5.29	49.66	72.99	−0.226	84.83	None/outside	nucl: 7, chlo: 6	Nucleus
TtTLP3	02G0003200	291–31.30	G (10.3%), S (9.3%), L (7.9%)	8.24	37.81	71.37	0.099	80.07	1/out to in	golg: 3.5, golg_plas: 3.5, vacu: 3, plas: 2.5, chlo: 2, extr: 2	Extracellular
TtTLP4	02G0007540	245–25.82	S (13.9%), G (10.2%), T (10.2%)	6.65	48.32	57.39	−0.129	87.76	None/outside	extr: 9, chlo: 4	Extracellular
TtTLP5	02G0018220	251–26.39	A (11.6%), G (11.2%), S (8.8%)	8.81	44.10	67.37	0.133	84.86	1/in to out	chlo: 9, extr: 2, vacu: 2	Extracellular
TtTLP6	03G0001120	247–26.44	G (10.9%), L (9.3%), P (8.9%)	7.88	50.49	66.76	−0.065	87.85	None/outside	extr: 6, chlo: 3, vacu: 2, nucl: 1, mito: 1	Cell wall
TtTLP7	03G0019340	659–73.67	L (10.3%), G (8.3%), V (7.1%)	8.47	33.15	84.43	−0.192	59.79	None/outside	chlo: 5, nucl: 4, cyto: 4	Chloroplast
TtTLP8	04G0001170	179–19.33	G (10.1%), L (8.9%), T (8.9%)	8.13	53.30	66.03	−0.040	78.77	None/outside	extr: 7, chlo: 2, vacu: 2, nucl: 1, mito: 1	Cell wall
TtTLP9	04G0015010	244–26.17	G (10.7%), L (7.8%), V (7.8%)	8.09	41.77	73.48	−0.015	83.20	1/in to out	chlo: 12, extr: 2	Extracellular
TtTLP10	04G0025610	226–23.58	G (11.9%), T (11.1%), S (8.4%)	4.68	29.98	50.53	−0.133	92.04	1/in to out	chlo: 5, extr: 4, vacu: 4	Extracellular
TtTLP11	04G0025620	227–24.02	G (11.9%), T (10.1%), P (8.4%)	8.40	19.66	54.58	−0.202	88.55	None/outside	chlo: 8, extr: 4, vacu: 2	Vacuole
TtTLP12	04G0025640	232–24.86	G (9.9%), T (9.5%), P (8.2%)	6.08	42.10	60.60	−0.314	88.36	None/outside	extr: 8, chlo: 4, vacu: 1	Extracellular
TtTLP13	06G0000860	384–39.82	G (13.5%), S (13.0%), P (11.5%)	4.80	56.96	57.92	−0.128	85.42	2/in to in	vacu: 4, golg_plas: 4, plas: 3.5, golg: 3.5, extr: 3	Extracellular
TtTLP14	06G0011700	244–26.12	G (10.7%), L (8.2%), C (7.4%), S (7.4%)	7.80	42.85	71.52	−0.029	81.97	None/outside	extr: 7, chlo: 3, nucl: 1, cyto: 1, vacu: 1	Extracellular
TtTLP15	06G0023190	227–23.67	G (12.3%), T (11.0%), S (8.4%)	4.68	28.01	51.19	−0.138	91.63	1/in to out	chlo: 6, vacu: 4, extr: 3	Extracellular
TtTLP16	06G0023200	227–24.00	G (11.9%), T (10.1%), P (8.8%)	8.40	19.01	54.58	−0.203	88.55	None/outside	chlo: 7, extr: 4, vacu: 2	Extracellular
TtTLP17	06G0023220	232–24.87	G (9.9%), T (9.9%), P (9.1%)	5.08	38.91	61.03	−0.284	89.66	None/outside	extr: 8, chlo: 4, vacu: 1	Vacuole
TtTLP18	07G0002190	274–29.60	S (11.7%), T (9.1%), G (8.0%)	5.14	46.68	68.36	−0.094	84.67	None/outside	nucl: 6, mito: 4, chlo: 3	Extracellular
TtTLP19	07G0015670	226–24.49	T (9.3%), G (8.8%), N (8.4%)	7.84	30.83	60.04	−0.188	85.84	None/outside	chlo: 14	Vacuole
TtTLP20	07G0019970	364–39.62	L (9.3%), A (8.8%), S (8.8%)	8.64	42.92	79.09	0.001	71.43	None/outside	chlo: 4, mito: 3, vacu: 3, extr: 2, nucl: 1	Cell wall
TtTLP21	07G0019980	355–36.81	G (13.0%), T (13.0%), S (11.5%)	4.31	31.77	57.21	−0.032	90.14	1/out to in	chlo: 4, extr: 4, vacu: 3, nucl: 2	Chloroplast
TtTLP22	09G0001980	332–34.63	S (12.0%), G (10.8%), T (10.5%)	4.70	38.64	64.70	−0.121	78.31	None/outside	chlo: 11, extr: 2	Extracellular
TtTLP23	09G0001990	340–35.38	S (14.7%), G (12.1%), T (10.0%)	4.73	47.62	55.18	0.010	83.82	2/out to out	extr: 5, golg: 3.5, golg_plas: 2.5, chlo: 2, vacu: 2	Extracellular
TtTLP24	10G0001950	410–45.74	L (10.5%), T (9.8%), S (8.8%)	9.28	39.04	83.17	−0.092	66.59	1/in to out	plas: 8.5, golg_plas: 5.5, golg: 1.5, chlo: 1, cyto: 1, vacu: 1	Chloroplast
TtTLP25	11G0010740	246–26.20	T (13.0%), P (8.5%), G (7.7%), L (7.7%)	6.66	37.24	70.16	0.080	81.71	None/outside	chlo: 14	Extracellular
TtTLP26	11G0017940	272–29.88	L (8.5%), P (7.4%), S (7.4%)	6.77	49.27	81.73	−0.024	75.37	1/in to out	chlo: 8, extr: 2, vacu: 2, nucl: 1	Chloroplast
TtTLP27	13G0000550	359–37.26	T (12.8%), G (12.5%), S (11.7%)	4.27	31.77	58.25	−0.025	86.91	1/out to in	chlo: 4, extr: 3, nucl: 2, vacu: 2, E.R.: 1.5, E.R._plas: 1.5	Extracellular
TtTLP28	13G0000560	317–33.62	L (10.7%), A (10.4%), G (9.5%)	8.27	37.09	83.47	0.268	72.56	1/out to in	chlo: 6, extr: 4, plas: 3	Cell wall
TtTLP29	14G0003240	290–31.12	G (10.3%), S (9.0%), L (7.6%), T (7.6%), V (7.6%)	7.35	35.08	70.59	0.108	77.59	2/in to in	golg: 3.5, golg_plas: 3.5, extr: 3, vacu: 3, plas: 2.5, chlo: 2	Extracellular
TtTLP30	14G0007370	825–91.27	L (9.5%), S (9.2%), G (7.5%)	8.58	51.43	81.22	−0.168	46.55	None/outside	nucl: 6, chlo: 4, mito: 2, plas: 1	Cytoplasm
TtTLP31	14G0013960	259–26.33	G (15.1%), S (11.6%), T (9.3%)	4.23	36.07	58.03	−0.020	84.94	1/in to out	chlo: 13	Extracellular
TtTLP32	14G0017910	251–26.39	A (12.4%), G (11.2%), S (8.4%)	9.01	42.22	69.32	0.100	86.45	None/outside	chlo: 10, mito: 3	Cell wall
TtTLP33	15G0017250	342–36.53	S (14.9%), G (12.6%), T (8.5%)	4.90	46.67	59.42	0.011	85.96	2/out to out	extr: 7, chlo: 3, vacu: 2, mito: 1	Extracellular
TtTLP34	15G0017260	334–34.89	S (12.3%), T (10.5%), G (10.2%)	4.64	38.09	61.41	−0.122	82.93	None/outside	nucl: 7.5, chlo: 5, cyto_nucl: 4.5	Extracellular
TtTLP35	16G0001800	346–35.10	G (14.2%), S (13.9%), T (12.7%)	4.51	43.80	52.57	−0.136	91.91	None/outside	extr: 5, chlo: 2, nucl: 2, plas: 2, vacu: 2	Chloroplast
TtTLP36	16G0005420	513–58.38	L (11.5%), G (7.2%), K (6.8%)	8.40	30.92	89.84	−0.248	44.64	None/outside	cyto: 12, nucl: 2	Chloroplast
TtTLP37	16G0019580	358–37.04	G (13.7%), S (11.2%), P (10.3%)	4.80	53.04	54.53	−0.219	87.71	1/in to out	extr: 9, vacu: 3, golg: 2	Extracellular

MW: molecular weight; PI: isoelectric point; II: instability index; AI: aliphatic index; GRAVY: grand average of hydropathicity. The molecular weight and isoelectric points of predicted TtTLPs were detected using the ExPASy proteomics server (https://web.expasy.org/protparam/, accessed on 20 November 2023). The contents of disordered amino acids (aa, %) in TtTLPs were calculated according to the online program Disordered by loops/coils definition from DisEMBL 1.5 (Intrinsic Protein Disorder Prediction, http://dis.embl.de/, accessed on 20 November 2023). The TMHMM Server 2.0 program (https://services.healthtech.dtu.dk/services/TMHMM-2.0/, accessed on 20 November 2023) was used to predict the transmembrane helices, and the topologies of TtTLPs were also performed with the 3D prediction by PHYRE^2^ (http://www.sbg.bio.ic.ac.uk/phyre2/html/page.cgi?id=index, accessed on 20 November 2023). For the subcellular localization prediction, the online programs WoLF_PSORT (https://www.genscript.com/wolf-psort.html, accessed on 20 November 2023) and Plant-PLoc (http://www.csbio.sjtu.edu.cn/bioinf/plant/, accessed on 20 November 2023) were used.

## Data Availability

Data are contained within the article.

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
