# Peer review of "Genome-Wide Identification and Expression Analyses of the Thaumatin-Like Protein Gene Family in Tetragonia tetragonoides (Pall.) Kuntze Reveal Their Functions in Abiotic Stress Responses"

_plants, 2024, doi:10.3390/plants13172355_

Round 1

Reviewer 1 Report

Comments and Suggestions for Authors

plants-3145194

Title: Genome-wide identification and expression analyses of the Thaumatin-like protein gene family in Tetragonia tetragonoides (Pall.) Kuntze reveal their functions in abiotic stress responses

This manuscript reports a comprehensive study on the Thaumatin-like protein gene family in Tetragonia tetragonoides, highlighting the identification of 37 TLP genes and their potential roles in stress responses in this species. This research is a significant contribution to the field, mainly due to its focus on halophyte species like Tetragonia tetragonoides, which has been underrepresented in the literature. Identifying TLPs in this species significantly enhances our understanding of plant stress responses. However, it is essential to note that further studies are necessary to explore the functional roles of these genes in vivo fully.

General comments: The paper is well-written, and the technical language is accurate and appropriate. However, there are a few instances where sentence structure needs to be simplified for clarity and ease of reading, particularly in the more complex sections of the results. Some minor grammatical errors in the abstract and discussion sections need correction.

The thorough methodology section includes appropriate bioinformatics tools for gene identification and expression analysis. RNA sequencing and qRT-PCR for validating expression patterns and yeast wild-type and metal-sensitive mutants for functional identification related to heavy-metal tolerance induced by TtTLPs are vital points to the study. The phylogenetic analysis and motif characterization add depth to the findings. The results are well-organized and provide clear insights into the distribution and expression patterns of TtTLP genes. 

Without detriment to the manuscript's accomplishment and value, this reviewer has some suggestions/comments /queries that are listed below:

Abstract: This section could briefly mention the main findings from the expression analyses to provide a more comprehensive overview.

Introduction:

The introduction could be enhanced by including more details on the specific challenges of salinity and drought stress in coastal plants before entering the thaumatin-like protein family description.

Line 47: distributed in place of "distributes"

Line 48: "belongs to halophyte" is probably not the best way to designate this species type. Please use "is a halophyte" instead.

M&Ms:

This section could include more details on the selection criteria for the TLP genes used in functional assays.

Results: 

Overall, the legends on the figures are too small for easy reading. Will this be fixed when setting the final printing?

In Figure 10, the legend of each graph is not always visible, and in the graph TtTLP12/17, the Y-axis values are not visible.

Some of the figures, particularly in the supplementary material, , are complex and could benefit from additional labelling or a more detailed legend to aid interpretation.

Discussion and Conclusions: 

The discussion appropriately ties the findings to the broader context of plant stress responses. The authors make convincing arguments about the role of TLPs in environmental adaptability. However, they could explore alternative explanations for the observed differential expression patterns, particularly about specific abiotic stresses.

The conclusions effectively summarize the study's contributions and potential applications. The suggestion to target TLPs to improve crop stress tolerance is particularly compelling. However, the authors might consider addressing how their findings could be further validated in field conditions, which would enhance the practical relevance of the study.

Final comments:

While the study provides significant insights, it raises several questions: Are the identified TLPs conserved across other halophytes, and if so, how do their expression patterns compare? Could other factors not accounted for in this study influence the expression of TLPs under abiotic stress?

Author Response

Reviewer 1

This manuscript reports a comprehensive study on the Thaumatin-like protein gene family in Tetragonia tetragonoides, highlighting the identification of 37 TLP genes and their potential roles in stress responses in this species. This research is a significant contribution to the field, mainly due to its focus on halophyte species like Tetragonia tetragonoides, which has been underrepresented in the literature. Identifying TLPs in this species significantly enhances our understanding of plant stress responses. However, it is essential to note that further studies are necessary to explore the functional roles of these genes in vivo fully.

General comments: The paper is well-written, and the technical language is accurate and appropriate. However, there are a few instances where sentence structure needs to be simplified for clarity and ease of reading, particularly in the more complex sections of the results. Some minor grammatical errors in the abstract and discussion sections need correction.

The thorough methodology section includes appropriate bioinformatics tools for gene identification and expression analysis. RNA sequencing and qRT-PCR for validating expression patterns and yeast wild-type and metal-sensitive mutants for functional identification related to heavy-metal tolerance induced by TtTLPs are vital points to the study. The phylogenetic analysis and motif characterization add depth to the findings. The results are well-organized and provide clear insights into the distribution and expression patterns of TtTLP genes.

Without detriment to the manuscript's accomplishment and value, this reviewer has some suggestions/comments /queries that are listed below:

Response: Great thanks for your suggestion and affirmation. We have incorporated changes that reflect the detailed suggestions you graciously provided, and we hope to meet with your approval. Revised portion are marked in red in the manuscript. The main corrections and the responds to your comments are listed in the follows.

Abstract:

This section could briefly mention the main findings from the expression analyses to provide a more comprehensive overview.

Response: We have added the sentence “The expression pattern changes of TtTLPs provided a more comprehensive overview of this gene family being involved in multiple abiotic stress responses.” in abstract part.

Introduction:

The introduction could be enhanced by including more details on the specific challenges of salinity and drought stress in coastal plants before entering the thaumatin-like protein family description.

Response: We have added the expression as “For tropical and subtropical coastal plants, their morphology has plastic to the environment change, including leaf succulent, salt bladder in epidermal cells, especially under the extreme salinity and drought stress challenges [3, 5]. ” in the first paragraph.

Line 47: distributed in place of "distributes"

Response: We have made this revision according to your suggestion.

Line 48: "belongs to halophyte" is probably not the best way to designate this species type. Please use "is a halophyte" instead.

Response: We have made this revision according to your suggestion.

M&Ms:

This section could include more details on the selection criteria for the TLP genes used in functional assays.

Response: Here we supplied the selection criteria for identification the TtTLP genes as “The T. tetragonoides genome was sequenced and submitted to NCBI database (NCBI accession number: JBBMRK000000000, unreleased). All T. tetragonoides proteins were identified with InterProscan (https://www.ebi.ac.uk/interpro/search/sequence/, accessed on 20 November 2023), and the conserved domains and motifs (e < 1 × 10 −5) were assessed. To characterized the TtTLP family members, the conserved TLP domain (pfam No. PF00314, PROSITE profile No. PS51367, or InterPro No. IPR001938) was searched as a model, and the protein sequences containing this domain were screened using HMM3.0 software. The domains were also confirmed using the NCBI CDD program (https://www.ncbi.nlm.nih.gov/structure/cdd, accessed on 20 November 2023). The T. tetragonoides proteins with TLP domain and corresponding genes were eventually identified as TtTLP family.” in “4.2. Identification and characterization of TtTLPs in T. tetragonoides” part.

Results:

Overall, the legends on the figures are too small for easy reading. Will this be fixed when setting the final printing?

Response: In the revised manuscript, we have adjusted all of the figures’ resolution ratio and improved the pictures’ quality.

In Figure 10, the legend of each graph is not always visible, and in the graph TtTLP12/17, the Y-axis values are not visible.

Response: Thanks for your reminder. We have corrected this error in our revised manuscript.

Some of the figures, particularly in the supplementary material, are complex and could benefit from additional labelling or a more detailed legend to aid interpretation.

Response: We have supplemented the figure legends of the supplementary material. We also adjust some figure legends in the body part of this manuscript.

Discussion and Conclusions:

The discussion appropriately ties the findings to the broader context of plant stress responses. The authors make convincing arguments about the role of TLPs in environmental adaptability. However, they could explore alternative explanations for the observed differential expression patterns, particularly about specific abiotic stresses.

The conclusions effectively summarize the study's contributions and potential applications. The suggestion to target TLPs to improve crop stress tolerance is particularly compelling. However, the authors might consider addressing how their findings could be further validated in field conditions, which would enhance the practical relevance of the study.

Response: We have emphasized the special morphology of Tetragonia tetragonoides (with salt bladder) and the importance for self-adjustment for maintaining osmotic pressure in vivo in discussion part as: “For example, T. tetragonoides is an inward secretohalophyte and can store excess salinity in the salt gland or salt bladder, which is widely distributed in the epidermal cells of T. tetragonoides leaves and stems. This salt isolation strategy is also an efficient mechanism of dealing with hypersaline and hypertonic environments [36, 37]. Coupled with the specialization of vegetative morphology, the intrinsic molecular mechanism of T. tetragonoides plants for habitat adaptation impacts the genome, especially functional genes. Therefore, it is quite necessary to understand the mechanisms that T. tetragonoides have adapted to extreme environmental abiotic stress, including drought, saline-alkaline, and high temperature stress, to complete its life cycle.”. Undoubtedly, the TLP/osmotin genes are critical acting factors for maintaining osmotic pressure and reducing salt ion toxicities, then raises the topic of this research.

Final comments:

While the study provides significant insights, it raises several questions: Are the identified TLPs conserved across other halophytes, and if so, how do their expression patterns compare? Could other factors not accounted for in this study influence the expression of TLPs under abiotic stress?

Promoter analyses.

Response: Whether form the gene sequences/protein motifs or from the gene expression regulation by promoter analysis, we haven’t found the specificity of TtTLPs compared with other halophytes' TLP families. The only interest point for TtTLPs involved in halophytes' ecological adaptability might be the gene amplification, just like in figure 2, the gene pairs with almost 100% similarity, including: TtTLP5/TtTLP32, TtTLP20/TtTLP28, TtTLP4/TtTLP30, TtTLP1/TtTLP25, TtTLP13/TtTLP37, TtTLP22/TtTLP34, TtTLP23/TtTLP33, TtTLP21/TtTLP27, TtTLP12/TtTLP17, TtTLP10/TtTLP15, TtTLP11/TtTLP16, TtTLP3/TtTLP29, TtTLP9/TtTLP14, TtTLP2/TtTLP26, TtTLP6/TtTLP8, and TtTLP7/TtTLP36. Further we're going to focus more on the functional identification of some specific TtTLP gene by transgenic assay, such as TtTLP11/TtTLP16, which will answer this question better.

Reviewer 2 Report

Comments and Suggestions for Authors

The article entitled " Genome-wide identification and expression analyses of the Thaumatin-like protein gene family in Tetragonia tetragonoides (Pall.) Kuntze reveal their functions in abiotic stress responses"   has tried to identify the Thaumatin-like protein genes for abiotic stress in Tetragonia tetragonoides. Some issues need to be concerned. 

1. How about the gene expression analysis in different tissues of T. tetragonoides seedlings under various stress challenges, where did the dataset come from?

2. Some figures were not clear, which should be high resolution, such as Figure 5.

3. Figure 6 and 7 had only one heatmap, maybe they could integrated together.

4. The content of heavy metal (HM) tolerance was inconsistent with the theme. The author should mainly focus on the  salt and drought tolerance.

5. RNA-seq data (40G) in this study should be shared into NCBI or others. 

Comments on the Quality of English Language

 Moderate editing of English language required.

Author Response

Reviewer 2

The article entitled " Genome-wide identification and expression analyses of the Thaumatin-like protein gene family in Tetragonia tetragonoides (Pall.) Kuntze reveal their functions in abiotic stress responses" has tried to identify the Thaumatin-like protein genes for abiotic stress in Tetragonia tetragonoides. Some issues need to be concerned.

  1. How about the gene expression analysis in different tissues of T. tetragonoidesseedlings under various stress challenges, where did the dataset come from?

Response: The whole RNA-seq data (40G) of the T. tetragonoides (40 G) will be analyzed and integrated into another report and submitted to NCBI soon.

  1. Some figures were not clear, which should be high resolution, such as Figure 5.

Response: Thanks for your reminder. We have adjusted all of the figures’ resolution ratio and improved the pictures’ quality in the revised manuscript.

  1. Figure 6 and 7 had only one heatmap, maybe they could integrated together.

Response: We have integrated the figures 6 and 7 as one figure (Figure 6) in the revised manuscript, and the other figures’ numbers were also adjusted accordingly.

  1. The content of heavy metal (HM) tolerance was inconsistent with the theme. The author should mainly focus on thesalt and drought tolerance.

Response: Thank you very much for your suggestions. Here we considered that T. tetragonoides is a seashore halophyte, and many halophyte species are natural heavy metal (HM) hyperaccumulators, as well as plant TLPs/osmotins being Cys-rich proteins, so we also detected the HM tolerance mediated by TtTLPs in this study. We also hope finding some HM-detoxification genes, such as plant TLPs, which are really involved in HM responses and accumulation. And also, considering the natural habitats of T. tetragonoides plants, the HM pollution is inevitable, the HM-responding research in T. tetragonoides is another research topic.

  1. RNA-seq data (40G) in this study should be shared into NCBI or others.

Response: The genome sequencing data of T. tetragonoides have been submitted to NCBI (NCBI accession number: JBBMRK000000000, unreleased). The RNA-seq data of TtTLP genes are listed in Supplementary material of this manuscript as Table S4 (FPKM values of TtTLPs for the RNA-seq assay of T. tetragonoides tissues in this study). The whole RNA-seq data (40G) of the T. tetragonoides (40 G) will be released soon.

  1. Comments on the Quality of English Language

 Moderate editing of English language required.

Response: English language editing of this manuscript has been made by Letpub.

Reviewer 3 Report

Comments and Suggestions for Authors

This manuscript is dedicated to the genome-wide identification and analysis of the expression patterns of the thaumatin-like protein family in Tetragonia tetragonoides (Pall.) Kuntze, as well as the determination of their role in response to various abiotic stresses.

A comprehensive multigenic family of thaumatin-like proteins was identified, and the localization of 37 members of this family on specific chromosomes was determined. The phylogenetic relationships among these proteins and their genes from different plant species were analyzed, as were the domain structures of these protein sequences. Additionally, the promoters for these genes were analyzed in order to gain a better understanding of their transcriptional regulation.This study was conducted through a significant number of practical experiments and methods, including RT-PCR and yeast transformations of coding regions of thaumatin-like protein genes. Experiments were performed to simulate abiotic stressors on plants and assess the level of expression of target genes. Additionally, yeast transformations were conducted to confirm the functions of these proteins, as well as abiotic stress experiments.

In this work, extensive research was conducted in bioinformatics, which was closely linked with practical experiments. This demonstrates the highest level of research in this area. The topic of identifying the molecular mechanisms behind plant responses to abiotic stress, as well as searching for genes and proteins that could potentially aid cultivated plants in adapting to a rapidly changing environment, continues to be relevant.

This study is novel and suitable for publication in a journal such as Plants.The findings of this research can contribute to a better understanding of the mechanisms by which plants adapt to adverse conditions, and can assist in the development of techniques to enhance their resistance to stressful factors.

The paper is well-organized, well-referenced, and written in a clear and concise manner. The references cited are primarily recent publications, and there are no excessive numbers of self-references. The paper is scientifically rigorous, and the experimental design is suitable for testing the hypotheses. The results are reproducible, and the figures and tables are appropriately presented. The data is accurately displayed, and it is supported by supplementary files. The statistical analysis presented is valid, with a sufficient number of biological and technical replicates. The conclusions drawn are consistent with the presented evidence and arguments.

**Questions:**

1. Based on the introduction, it is unclear whether these proteins are specific to all higher plants, or if they are found in different plant families that are unrelated, and if there are differences in the number of gene families among different plant species.

2. For Figure 2, I would like to see an outer group and a ruler, as well as the designation of colors for the domains in the legend.

3. Generally, I recommend that all images be made larger, as the axis labels are difficult to read.

4. Regarding lines 484-491, can you please explain how you determined that the genes represented in pairs are duplications rather than allelic variants?

5. Why was yeast used for transforming to determine the function of tlp family proteins, instead of Arabidopsis, which is a commonly used model organism for studying plant molecular physiology?

Author Response

Reviewer 3

This manuscript is dedicated to the genome-wide identification and analysis of the expression patterns of the thaumatin-like protein family in Tetragonia tetragonoides (Pall.) Kuntze, as well as the determination of their role in response to various abiotic stresses.

A comprehensive multigenic family of thaumatin-like proteins was identified, and the localization of 37 members of this family on specific chromosomes was determined. The phylogenetic relationships among these proteins and their genes from different plant species were analyzed, as were the domain structures of these protein sequences. Additionally, the promoters for these genes were analyzed in order to gain a better understanding of their transcriptional regulation.This study was conducted through a significant number of practical experiments and methods, including qRT-PCR and yeast transformations of coding regions of thaumatin-like protein genes. Experiments were performed to simulate abiotic stressors on plants and assess the level of expression of target genes. Additionally, yeast transformations were conducted to confirm the functions of these proteins, as well as abiotic stress experiments.

In this work, extensive research was conducted in bioinformatics, which was closely linked with practical experiments. This demonstrates the highest level of research in this area. The topic of identifying the molecular mechanisms behind plant responses to abiotic stress, as well as searching for genes and proteins that could potentially aid cultivated plants in adapting to a rapidly changing environment, continues to be relevant.

This study is novel and suitable for publication in a journal such as Plants.The findings of this research can contribute to a better understanding of the mechanisms by which plants adapt to adverse conditions, and can assist in the development of techniques to enhance their resistance to stressful factors.

The paper is well-organized, well-referenced, and written in a clear and concise manner. The references cited are primarily recent publications, and there are no excessive numbers of self-references. The paper is scientifically rigorous, and the experimental design is suitable for testing the hypotheses. The results are reproducible, and the figures and tables are appropriately presented. The data is accurately displayed, and it is supported by supplementary files. The statistical analysis presented is valid, with a sufficient number of biological and technical replicates. The conclusions drawn are consistent with the presented evidence and arguments.

Response:

  Thank you for your comments and approvals concerning our study and manuscript. We would like very much to dig out some functional stress-tolerance genes from the special habitat plant species, such as Tetragonia tetragonoides.

  1. Based on the introduction, it is unclear whether these proteins are specific to all higher plants, or if they are found in different plant families that are unrelated, and if there are differences in the number of gene familiesamong different plant species.

Response: We mentioned the TLP family gene numbers of several plant species in introduction part as: “Melon (Cucumis melo), cotton (Gossypium barbadense), strawberry (Fragaria × ananassa, 2n = 8x = 56), and garlic (Allium sativum) genomes contain 29, 90, 76, and 32 TLPs, respectively [23–26], and they all play important roles in developmental processes and diverse stress condition responses, especially in disease resistance.”. But, evidently, the gene numbers of families are related with the size and ploidy of plant genomes, so it needs a concrete analysis of specific situations of specific gene family.

  1. For Figure 2, I would like to see an outer group and a ruler,as well as the designation of colors for the domains in the legend.

Response: Thanks for your kindly reminder. We have corrected the picture typographical error in Figure 2, and we also added the ruler.

  1. Generally, I recommend that all imagesbe made larger, as the axis labels are difficult to read.

Response: In the revised manuscript, we have adjusted all of the figures’ resolution ratio and improved the pictures’ quality.

  1. Regarding lines 484-491, can you please explain how you determined that the genes represented in pairs are duplications rather than allelic variants?

Response: Here we can not distinguish between gene duplication and allelic variants, so we just based on the phylogenetic analysis of TtTLP proteins (Figure 2), this gene amplification in a family was considered as gene duplication, which might also include allelic variants, especially in this special habitat plant species, T. tetragonoides.

  1. Why was yeastused for transforming to determine the function of TLPfamily proteins, instead of Arabidopsis, which is a commonly used model organism for studying plant molecular physiology?

Response: The yeast expression system has the advantages that many other transgenic systems don't include, for instance, highly efficient, simple operation, and easily detected. For initial functional identification of some genes, we can preliminary acquire the specific phenotype when expressed in yeast, and then carry out functional identification in a deep-going way by plant transgenic assays. Here in this research, we will pay more attention to TtTLP11/TtTLP16 gene pair, which will be over-expressed in Arabidopsis and study the transgenic plants’ molecular physiology.
